# LIBRA-EMO: A LARGE DATASET FOR MULTIMODAL FINE-GRAINED NEGATIVE EMOTION DETECTION

## ABSTRACT

The recognition of negative emotions is pivotal in numerous real-world applications, including public opinion analysis, customer service, emotional attribution, and emotional support systems, where these emotions manifest with fine-grained characteristics. However, current models struggle with fine-grained negative emotion recognition tasks due to the limited granularity in existing multimodal emotion recognition datasets. To address this, we refine coarse-grained emotion categories, expanding negative emotions from conventional 4-5 types to 8 specific categories. Based on this refined taxonomy, we construct **Libra-Emo**, a comprehensive dataset for multimodal fine-grained negative emotion detection. It comprises **Libra-Emo Trainset** for model development and **Libra-Emo Bench** for evaluation, collectively containing 62,267 video samples annotated through a novel human-machine collaborative active learning strategy, surpassing existing datasets in both granularity and scale. We present extensive experimental results from zero-shot evaluations using Libra-Emo Bench and instruction-tuning experiments with Libra-Emo Trainset on leading Multimodal Large Language Models (MLLMs). Our findings demonstrate that while current MLLMs exhibit limited proficiency in fine-grained negative emotion detection, models fine-tuned on Libra-Emo Trainset show substantial performance improvements that generalize effectively to out-of-domain evaluations. This work addresses critical limitations in existing multimodal emotion recognition datasets regarding emotion category granularity and representation of negative emotions, thus advancing research in fine-grained emotional analysis. The dataset and models will be fully open-sourced.

## 1 INTRODUCTION

Emotion recognition (Poria et al., 2019b; Khare et al., 2024; Wang et al., 2022) has become a crucial component in human-computer interaction, public opinion monitoring, and intelligent customer service, where understanding human emotions enables more empathetic and effective communication.

Benefiting from the rapid development of Multimodal Large Language Models (MLLMs) (OpenAI, 2024; Google, 2024; Anthropic, 2025), various multimodal emotion recognition systems such as EmoCLIP (Jiang et al., 2023), FARCER (Lei et al., 2024), and Emotion-LLaMA (Cheng et al., 2024) have been proposed, aiming to analyze emotional states in videos and images by integrating multimodal information including visual and linguistic cues. Although these approaches have achieved progress in enhancing emotion understanding, they still face the following limitations:

- **Limited granularity in emotion categories.** Existing multimodal emotion detection datasets are based on the six coarse-grained emotion categories defined by Paul Ekman (Ekman et al., 2013) (see Table 1). However, we observe that existing MLLMs exhibit suboptimal performance in recognizing coarse-grained emotion categories (see Table 2). We partially attribute this to the inherent ambiguity and overlapping boundaries associated with coarse-grained categories.

- **Insufficient attention to negative emotions.** The recognition of negative emotions plays a vital role in numerous real-world applications, such as public opinion analysis, customer service, emotion attribution, and emotional support systems (Ham et al., 2023; Guo et al., 2024). Accurate identification of negative emotions facilitates more precise and effective

response strategies. Nevertheless, existing multimodal emotion detection datasets provide insufficient support for negative emotions (see Table 1).

These limitations highlight the urgent need for dedicated datasets and evaluation frameworks to ensure effective detection and analysis of fine-grained negative emotions in multimodal content.

To address these challenges, we first refine the existing coarse-grained emotion categories, with a particular focus on negative emotions. The mapping between coarse-grained and fine-grained emotion categories is provided in Appendix F. We find that using fine-grained emotion categories significantly improves the model's emotion recognition capabilities (see Table 2), suggesting that fine-grained labels reduce ambiguity and mitigate misclassifications caused by emotional vagueness. Building on this finding, we propose the **Libra-Emo**, an advanced multimodal fine-grained negative emotion detection dataset specifically designed for comprehensive emotion analysis in multimodal content. Specifically, Libra-Emo employs a carefully designed data collection process and a human-machine collaborative active learning annotation strategy to construct a diverse dataset comprising 13 emotion categories, including 8 distinct negative emotions. The training subset, **Libra-Emo Trainset**, contains 61,625 samples, while the evaluation subset, **Libra-Emo Bench**, consists of 642 samples, providing a solid foundation for research in multimodal fine-grained negative emotion detection.

To validate the effectiveness of Libra-Emo, we conduct comprehensive zero-shot evaluations using Libra-Emo Bench and instruction-tuning experiments with Libra-Emo Trainset on leading MLLMs. Experimental results demonstrate that while current MLLMs exhibit limited proficiency in fine-grained negative emotion detection, models fine-tuned on Libra-Emo Trainset show substantial performance improvements. Moreover, experiments demonstrate that the performance improvements brought by Libra-Emo Trainset generalize to the out-of-domain test set DFEW (Jiang et al., 2020). These results establish Libra-Emo as a robust framework for advancing fine-grained negative emotion detection in multimodal content, thereby facilitating more nuanced emotion understanding across diverse applications.

Our contributions can be summarized as follows:

> **Libra-Emo Taxonomy**: A novel emotion classification framework that expands traditional categories into 13 distinct emotional states with particular emphasis on 8 fine-grained negative emotions, enabling more nuanced emotional analysis than existing taxonomies.
> **Libra-Emo Trainset**: A large-scale multimodal fine-grained negative emotion detection dataset comprising 61,625 annotated video samples, surpassing existing datasets in both granularity and volume.
> **Libra-Emo Bench**: A comprehensive benchmark for assessing the performance of multimodal models in fine-grained negative emotion recognition, covering a wide range of scenarios and providing a valuable resource for the research community.
> **Libra-Emo Model**: We fine-tune a series of leading MLLMs on Libra-Emo Trainset, significantly enhancing their performance in negative emotion recognition tasks and demonstrating the value of specialized datasets.

Table 1: Comparison with other video emotion detection datasets.

| Dataset | # Emo. | # Neg. Emo. | # Samps. |
|---|---|---|---|
| IEMOCAP (Busso et al., 2008) | 9 | 5 | 10,039 |
| CREMA-D (Cao et al., 2014) | 6 | 4 | 7,442 |
| MELD (Poria et al., 2019a) | 7 | 4 | 13,000 |
| CAER (Lee et al., 2019) | 7 | 4 | 13,201 |
| CMU-MOSEI (Zadeh et al., 2018b) | 7 | 4 | 23,453 |
| **Libra-Emo (Ours)** | **13** | **8** | **62,267** |

Table 2: Refining emotion granularity on Libra-Emo Bench.

| Model | CLS | ACC | F1 |
|---|---|---|---|
| Gemini-2.0-Flash (Google, 2024) | 7-CLS | 53.89 | 53.65 |
| | 13-CLS → 7-CLS | **58.88** | **59.04** |
| GPT-4o (OpenAI, 2024) | 7-CLS | 40.97 | 43.07 |
| | 13-CLS → 7-CLS | **53.58** | **53.03** |
| Claude-3.7-Sonnet (Anthropic, 2025) | 7-CLS | 42.37 | 41.64 |
| | 13-CLS → 7-CLS | **54.98** | **54.28** |

## 2 LIBRA-EMO CONSTRUCTION

Figure 1 provides an overview of the construction process of Libra-Emo, which mainly includes emotion categories definition, video collection and processing, and emotion annotation.

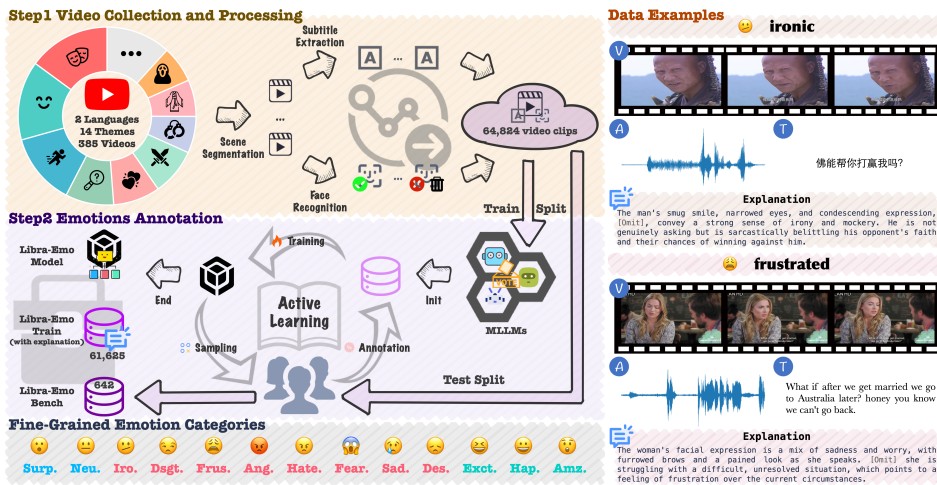

Figure 1: Overview of the dataset construction process for Libra-Emo.

## 2.1 EMOTION CATEGORIES DEFINITION

The first step in constructing Libra-Emo is to establish a comprehensive and fine-grained emotion taxonomy. Drawing upon psychological research and practical applications [1], we expand Paul Ekman's 6 basic emotions (Ekman et al., 2013) into a more granular set of 13 distinct categories, with a particular emphasis on negative emotions, which are divided into 8 specific types. Detailed definitions can be found in Table 3. This fine-grained categorization, particularly for negative emotions, fills a critical gap in existing datasets and enables more nuanced emotion recognition.

## 2.2 VIDEO COLLECTION AND PROCESSING

**Video Collection.** To ensure diversity and relevance, the source videos are collected from a wide range of TV shows and movies on YouTube [2], all of which are licensed under Creative Commons. These videos feature English and Chinese and cover 14 themes, including drama, comedy, romance, action, adventure, sci-fi, fantasy, horror, thriller, crime, war, family, school, and historical drama. This selection aims to provide a balanced representation of different cultural backgrounds and production styles. A total of 385 source videos covering 2 languages and 14 themes are collected, as outlined in Appendix A, thereby ensuring that the resulting emotion samples encompass a wide range of contexts and expressive styles.

**Video Processing.** The collected videos are processed to extract meaningful segments and relevant information:

1. **Scene Segmentation**: We utilize the *scenedetect* tool [3] to identify natural scene boundaries within the source videos and retain video clips that are longer than 3 seconds and at most 10 seconds, thereby creating coherent, emotion-bearing clips. Each clip is then analyzed for its primary emotional content.

2. **Subtitle Extraction**: For the collected raw videos, we download subtitle files in the corresponding languages and obtain the subtitles for each clip according to the timestamps of the segments, enabling subsequent training and testing. Clips without subtitles are filtered out.

3. **Face Recognition**: We employ the *face_recognition* tool [4] to perform facial detection on each video clip, sampling three frames per second and retaining clips in which faces appear in more than 99% of the frames.

---

[1]https://en.wikipedia.org/wiki/Emotion_classification
[2]https://www.youtube.com
[3]https://pypi.org/project/scenedetect
[4]https://github.com/ageitgey/face_recognition

Table 3: Emotion categories in Libra-Emo.

| Type | Category | Definition |
|---|---|---|
| **Positive** | Excited | A high-energy, positive state marked by eagerness, anticipation, and enthusiasm. |
| | Happy | A general sense of contentment, joy, or life satisfaction, often calm and sustained. |
| | Amazed | A strong and lasting sense of wonder or astonishment, often triggered by something extraordinary or impressive. |
| **Neutral** | Surprised | An immediate emotional response to an unexpected event, marked by sudden awareness or shock. |
| | Neutral | An emotionally unmarked state, indicating neither positive nor negative affect. |
| **Negative** | Ironic | A sarcastic or mocking emotional state, often marked by indirect critique or contradiction. |
| | Disgusted | A visceral reaction of revulsion or strong disapproval, often in response to something morally or physically offensive. |
| | Frustrated | A state of tension, irritation, and dissatisfaction resulting from obstacles that prevent achieving goals or expectations. |
| | Angry | A reactive emotion involving displeasure, irritation, or aggression, usually in response to perceived wrongs or injustice. |
| | Hateful | A persistent, intense hostility or contempt directed toward a person, group, or idea, often associated with a desire for harm. |
| | Fearful | A defensive emotion involving anxiety or dread, triggered by perceived threats or uncertainty. |
| | Sad | A low-energy emotion characterized by feelings of loss, disappointment, or helplessness. |
| | Despairful | A profound sense of hopelessness and helplessness, often accompanied by emotional distress and loss of purpose. |

Through this process, we generate 64,824 candidate samples containing visual, auditory, and textual information, with an average video length of 5.0 s and an average face ratio of 99.9%, creating a rich multimodal foundation for annotation.

## 2.3 EMOTION ANNOTATION

Fine-grained emotion labels pose significant challenges for annotation. To obtain large-scale annotated data while balancing labor costs and label quality, we employ an active learning strategy that combines model predictions with human annotation.

**Libra-Emo Bench**: We first sample a batch of data from the candidate samples for manual annotation to construct our test set, Libra-Emo Bench. The developed annotation tool is presented in Appendix G. Each sample is annotated by 8 individuals, with a voting threshold set to 4, meaning the annotation is considered successful when at least 4 annotators agree on the label. For samples that fail to reach consensus, the final annotation is determined through detailed discussions among multiple annotators. The final Libra-Emo Bench consists of 642 samples, with the category distribution shown in Figure 2.

**Active Learning Strategy for Trainset Annotation**: Model-based annotation can reduce costs while maintaining high consistency (Gilardi et al., 2023; Tan et al., 2024). We first consider using multiple MLLMs to vote for labeling the Libra-Emo Trainset. However, experimental results on a sampled test set from Libra-Emo Bench (Table 4) indicate that the current leading MLLMs do not perform well in fine-grained emotion recognition for videos. Therefore, we adopt a human-machine collaborative active learning strategy for training set annotation (Tharwat & Schenck, 2023; Li et al., 2024), aiming to maximize dataset quality while minimizing labor costs.

The algorithm for the annotation process is in Appendix D, with the textual description as follows:

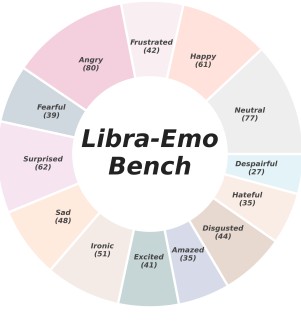

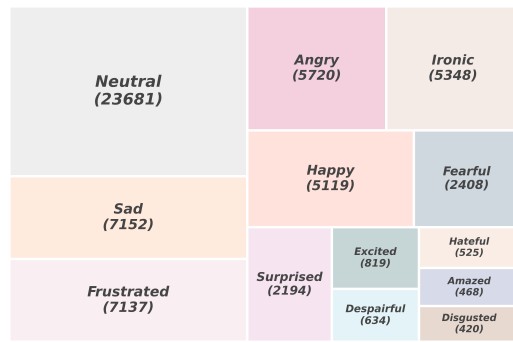

Figure 2: Libra-Emo Bench class distribution.     Figure 3: Libra-Emo Trainset class distribution.

1. **Initial Labeling**: We select Gemini-2.0-Flash (Google, 2024), GPT-4o (OpenAI, 2024), and Claude-3.7-Sonnet (Anthropic, 2025) as our voting models for initial labeling of the training set based on the voting performance shown in Table 4. For each sample in the unlabeled dataset, the three models independently predict a label. The prompt used for annotation can be found in Appendix B. If at least two models agree, the majority vote becomes the label and is added to the labeled dataset. Otherwise, the sample is annotated manually to determine the initial label.

2. **Iterative Label Refinement**: (1) Model Training: Train the model using the currently labeled dataset. (2) Sample Selection: The model predicts new labels for all labeled samples, and selects the samples where the new predictions differ from the current labels for human annotation. (3) Human Annotation: Annotators label the selected samples. Each sample is annotated by 4 individuals, with a voting threshold set to 2.

3. **Final Output**: The iteration stops when the model's performance reaches a plateau. The number of iterative rounds used for the Libra-Emo Trainset is 2. The final output is the trained model and the labeled dataset.

**Synthesizing Explanations Consistent with Labels**: Studies (Menon et al., 2022; Ferraretto et al., 2023; Zhang et al., 2024b) have demonstrated that natural language explanations have a positive impact on LLM classification tasks. We use Gemini-2.0-Flash to synthesize label-consistent explanations on the annotated dataset to enhance the accuracy of the emotion recognition model. The prompt for explanation synthesis can be found in Appendix B.

A detailed description of the construction process of the Libra-Emo Trainset is provided in Appendix C. The final Libra-Emo Trainset contains 61,625 samples, with the category distribution shown in Figure 3. Training examples can be found in Figure 1 and Appendix H. The composition distribution of the complete Libra-Emo is shown in Table 5.

Table 4: The performance of leading MLLMs on Libra-Emo Bench.

Table 5: Libra-Emo composition distribution. For more details, see Appendix F.

| Model | Acc | F1 |
|---|---|---|
| Gemini-2.0-Flash (Google, 2024) | 47.82 | 47.66 |
| GPT-4o (OpenAI, 2024) | 43.30 | 41.74 |
| Claude-3.7-Sonnet (Anthropic, 2025) | 41.74 | 37.26 |
| Vote | **50.93** | **50.85** |

| Emotion Type | Quantity | | |
|---|---|---|---|
| | Trainset | Bench | Total |
| **Positive** | 6,406 | 137 | 6,543 |
| **Neutral** | 25,875 | 139 | 26,014 |
| **Negative** | 29,344 | 366 | 29,710 |
| **Total** | 61,625 | 642 | 62,267 |

## 3 EXPERIMENTS

In this section, we present the experimental setup and results used to evaluate the performance of multimodal models on the Libra-Emo. We first describe the experimental settings, including model selection and evaluation metrics, followed by a comparison with baseline approaches. Then, we provide the main results, highlighting key insights from the evaluation.

## 3.1 Experimental Settings

**Model Selection**  To comprehensively evaluate fine-grained negative emotion recognition capabilities, we benchmark several leading multimodal large language models (MLLMs). Our selection encompasses diverse, recent, and publicly available MLLMs that represent varied architectural approaches and parameter scales, including LLaVA-Video-7B-Qwen2 (Zhang et al., 2024a), Qwen2.5-VL-7B (Team, 2025), Phi-3.5-vision-instruct (4.2B) (Abdin et al., 2024), MiniCPM-o 2.6 (8B) (Yao et al., 2024), Qwen-2.5-Omni-7B (Xu et al., 2025), and InternVL-2.5 series (1B-8B) (Chen et al., 2024). Detailed model descriptions are provided in Appendix E.

**Fine-tuning Settings**  We conduct experiments on Qwen-2.5-Omni-7B (Xu et al., 2025) and InternVL-2.5 series (1B-8B) (Chen et al., 2024). To validate the effectiveness of the dataset, we employ consistent fine-tuning and video processing configurations across all experiments. The models train for one epoch using the AdamW optimizer with a learning rate of 3e-4 (cosine decay following a linear warm-up) and a weight decay of 0.01. The prompt used for fine-tuning, which integrates video with subtitle text, is provided in Appendix B. Further details on hyperparameters are available in Appendix E.

**Evaluation Metrics**  **1. Libra-Emo Bench**: We report Accuracy, Macro-F1, and Weighted-F1 for both all emotions and negative emotions (considering only samples with ground truth labeled as negative emotions). **2. Out-of-Domain Test Set DFEW (Jiang et al., 2020)**: We use standard evaluation metrics UAR (Unweighted Average Recall) and WAR (Weighted Average Recall) to evaluate zero-shot inference performance on the set_1 collection of 2,341 samples.

**Baselines**  To comprehensively evaluate the effectiveness of models fine-tuned on Libra-Emo Trainset, the baseline approaches are divided into two categories: **Zero-shot MLLMs** (leading multimodal models used without any fine-tuning to assess their inherent emotion recognition capabilities) and **Existing Emotion Recognition Models** (specialized models trained on previous emotion datasets).

## 3.2 Libra-Emo Bench Evaluation

The experimental results presented in Table 6 reveal several important findings:

**Performance disparity between closed-source and open-source models**: Closed-source models outperform their open-source counterparts in zero-shot settings. Gemini-2.0-Flash achieves the highest accuracy (47.82%) and F1 scores (47.55% macro-F1, 47.66% weighted-F1) for all emotions, demonstrating strong performance in negative emotion recognition (49.18% accuracy, 48.30% weighted-F1). However, after fine-tuning on the Libra-Emo Trainset, open-source models substantially narrow this gap: Libra-Emo-Omni-7B achieves 51.56% accuracy across all emotions, while Libra-Emo-8B achieves 50.55% accuracy on negative emotions, significantly surpassing the closed-source models.

**Dataset effectiveness**: Compared with zero-shot models, fine-tuning on the Libra-Emo Trainset significantly improves the performance of models across different architectures and scales, validating the dataset's quality and usefulness. The most dramatic improvements are observed in smaller models, with Libra-Emo-1B showing a 17.91% increase in overall accuracy and Libra-Emo-2B exhibiting a 22.95% increase in negative emotion accuracy. These substantial gains highlight the efficacy of specialized training for fine-grained emotion recognition tasks.

**Impact of different modalities**: The results of different modalities on the same model demonstrate the performance improvement brought by incorporating modalities. Gemini-2.0-Flash achieves only 40.19% and 30.69% accuracy on visual-only and audio-only, respectively. The visual-audio (V,A) and visual-text (V,T) modalities show slight improvements, reaching 42.37% and 46.57%, respectively, while using all modalities (V,A,T) achieves the highest accuracy of 47.82%. Notably, after fine-tuning on the Libra-Emo training set, the three-modality model Libra-Emo-Omni-7B reaches an overall accuracy of 51.56% and a negative-emotion accuracy of 49.45%, both exceeding Gemini-2.0-Flash. These results highlight the significance of multimodal fusion and underscore the value of Libra-Emo for multimodal emotion recognition.

Table 6: Performance Comparison of MLLMs on **Libra-Emo Bench**, showing the Accuracy and F1 scores for all emotions and negative emotions. The modalities column indicates the content available during reasoning (V: visual, A: auditory, T: textual). All values are in percentages (%).

| Model | Modalities | All Emotions (13 Classes) | | | Negative Emotions (8 Classes) | | |
|---|---|---|---|---|---|---|---|
| | | Accuracy | Macro-F1 | Weighted-F1 | Accuracy | Macro-F1 | Weighted-F1 |
| *Closed-Source Models* | | | | | | | |
| Gemini-2.0-Flash (Google, 2024) | V | 40.19 | 38.08 | 39.11 | 36.07 | 37.83 | 37.48 |
| Gemini-2.0-Flash (Google, 2024) | A | 30.69 | 27.56 | 29.47 | 27.87 | 24.15 | 26.96 |
| Gemini-2.0-Flash (Google, 2024) | V,A | 42.37 | 41.44 | 41.82 | 41.26 | 41.12 | 40.55 |
| Gemini-2.0-Flash (Google, 2024) | V,T | 46.57 | 45.25 | 45.42 | 48.36 | 48.80 | 47.60 |
| Gemini-2.0-Flash (Google, 2024) | V,A,T | 47.82 | 47.55 | 47.66 | 49.18 | 48.83 | 48.30 |
| GPT-4o (OpenAI, 2024) | V,T | 43.30 | 44.47 | 41.74 | 45.90 | 46.91 | 45.00 |
| Claude-3.7-Sonnet (Anthropic, 2025) | V,T | 41.74 | 33.70 | 37.26 | 36.89 | 27.09 | 30.81 |
| *Open-Source Models* | | | | | | | |
| LLaVA-Video-7B-Qwen2 (Zhang et al., 2024a) | V,T | 19.63 | 13.28 | 15.24 | 7.92 | 9.98 | 11.30 |
| Qwen2.5-VL-7B (Team, 2025) | V,T | 38.32 | 33.99 | 34.70 | 35.25 | 29.47 | 31.03 |
| Phi-3.5-vision-instruct (4.2B) (Abdin et al., 2024) | V,T | 24.61 | 19.37 | 20.40 | 13.39 | 15.59 | 15.12 |
| MiniCPM-o 2.6 (8B) (Yao et al., 2024) | V,A,T | 19.78 | 14.78 | 15.87 | 17.49 | 11.45 | 14.41 |
| Qwen-2.5-Omni-7B (Xu et al., 2025) | V | 30.22 | 25.72 | 26.39 | 23.77 | 20.37 | 21.09 |
| Qwen-2.5-Omni-7B (Xu et al., 2025) | V,A | 35.20 | 30.72 | 30.61 | 30.33 | 27.62 | 28.32 |
| Qwen-2.5-Omni-7B (Xu et al., 2025) | V,A,T | 38.94 | 34.46 | 34.64 | 35.79 | 33.17 | 34.14 |
| InternVL-2.5-1B (Chen et al., 2024) | V,T | 21.50 | 13.19 | 15.93 | 16.94 | 9.32 | 11.86 |
| InternVL-2.5-2B (Chen et al., 2024) | V,T | 19.47 | 11.12 | 13.53 | 18.58 | 9.61 | 12.09 |
| InternVL-2.5-4B (Chen et al., 2024) | V,T | 32.40 | 27.42 | 26.73 | 25.68 | 25.53 | 26.67 |
| InternVL-2.5-8B (Chen et al., 2024) | V,T | 36.45 | 32.50 | 33.84 | 36.61 | 32.29 | 34.39 |
| *Fine-Tuned on Libra-Emo Trainset* | | | | | | | |
| Libra-Emo-Omni-7B (Ours) | V | 39.88 | 38.83 | 39.12 | 33.61 | 36.58 | 36.47 |
| Libra-Emo-Omni-7B (Ours) | V,A | 45.79 | 44.11 | 45.01 | 39.62 | 41.00 | 42.01 |
| Libra-Emo-Omni-7B (Ours) | V,A,T | **51.56** | 50.83 | **51.08** | 49.45 | 49.30 | 49.28 |
| Libra-Emo-1B (Ours) | V,T | 39.41 | 36.50 | 38.02 | 34.97 | 31.94 | 33.23 |
| Libra-Emo-2B (Ours) | V,T | 43.61 | 40.66 | 42.19 | 41.53 | 37.48 | 38.82 |
| Libra-Emo-4B (Ours) | V,T | 44.39 | 40.61 | 42.27 | 41.53 | 38.49 | 39.73 |
| Libra-Emo-8B (Ours) | V,T | 51.25 | **51.40** | 51.07 | **50.55** | **50.20** | **49.79** |

These findings indicate that while leading MLLMs possess some inherent capability for emotion recognition, they significantly underperform on fine-grained negative emotion detection without specialized training. The Libra-Emo Trainset effectively addresses this limitation, enabling substantial performance improvements across diverse model architectures through targeted fine-tuning.

## 3.3 OUT-OF-DOMAIN TEST SET EVALUATION

Table 7 presents zero-shot performance on the DFEW dataset (Jiang et al., 2020). Libra-Emo-8B achieves the highest overall metrics (52.55% UAR, 59.85% WAR) among all models, demonstrating strong generalization capabilities. Most notably, our models excel at recognizing negative emotions, particularly sad (82.59%), angry (71.26%), and fearful (66.30%) categories, outperforming specialized emotion recognition models like Emotion-LLaMA (Cheng et al., 2024) in these areas. These results validate the effectiveness of Libra-Emo Trainset for fine-grained negative emotion recognition across domains and underscore its practical value for emotion monitoring applications.

Table 7: Performance Comparison on DFEW in Zero-Shot Setting. All values are in percentages (%).

| Models | Happy | Surprised | Neutral | Sad | Angry | Disgusted | Fearful | UAR | WAR |
|---|---|---|---|---|---|---|---|---|---|
| Video-LLaVA (Lin et al., 2023) | 51.94 | 0.00 | 29.78 | 39.84 | 58.85 | 0.00 | 2.76 | 26.17 | 35.24 |
| Video-Llama (Zhang et al., 2023) | 20.25 | 4.76 | 80.15 | 67.55 | 5.29 | 0.00 | 9.39 | 26.77 | 35.75 |
| GPT-4V (Lian et al., 2024) | 62.35 | 32.19 | 56.18 | 70.45 | 50.69 | 10.34 | 51.11 | 47.69 | 54.85 |
| Emotion-LLaMA (Cheng et al., 2024) | **71.98** | 33.67 | 61.99 | 76.25 | **71.95** | 0.00 | 3.31 | 45.59 | 59.37 |
| Libra-Emo-Omni-7B (Ours) | 57.26 | **44.90** | 51.87 | 66.75 | 64.14 | 6.90 | **66.30** | 51.16 | 57.37 |
| Libra-Emo-8B (Ours) | 62.78 | 42.86 | 45.13 | **82.59** | 71.26 | 6.90 | 56.35 | **52.55** | **59.85** |

# 4  ABLATION STUDIES

In this section, we evaluate key design choices in the Libra-Emo framework to understand their impact on performance. Unless otherwise noted, all ablation experiments are conducted using the InternVL-2.5-8B model for consistency.

## 4.1  IMPACT OF ACTIVE LEARNING AND EXPLANATION

Table 8 demonstrates the effectiveness of our active learning strategy. As active learning progresses from Round 0 to Round 2, performance metrics steadily improve across the Libra-Emo Bench. The overall accuracy on Libra-Emo Bench increases from 46.26% to 50.00%, while negative emotion accuracy rises from 44.54% to 48.91%. Incorporating explanations in Round 2 provides a substantial performance boost, with overall accuracy reaching 51.25% (+1.25%) and negative emotion accuracy improving to 50.55% (+1.64%). These results validate our dataset construction methodology, demonstrating that combining iterative active learning with explanatory annotations significantly enhances emotion recognition capabilities, particularly in distinguishing fine-grained negative emotions.

Table 8: Ablation study on active learning and explanation. All values are in percentages (%).

| Activate Learning Round | Explanation | All Emotions (13 Classes) | | | Negative Emotions (8 Classes) | | |
|---|---|---|---|---|---|---|---|
| | | Accuracy | Macro-F1 | Weighted-F1 | Accuracy | Macro-F1 | Weighted-F1 |
| Round 0 | ✗ | 46.26 | 43.91 | 45.15 | 44.54 | 42.10 | 43.51 |
| Round 1 | ✗ | 48.44 | 46.76 | 47.55 | 46.99 | 44.71 | 45.36 |
| Round 2 | ✗ | 50.00 | 48.51 | 49.29 | 48.91 | 46.89 | 47.27 |
| Round 2 | ✔ | **51.25** | **51.40** | **51.07** | **50.55** | **50.20** | **49.79** |

## 4.2  IMPACT OF DATASET SIZE

Figure 4 illustrates the impact of training data scale on model performance. With only 12.5% of the data, the model already achieves substantial gains in overall accuracy (43.93%, +7.48%) and negative emotion accuracy (42.90%, +6.29%). As the dataset size increases, performance continues to improve steadily, ultimately reaching 51.25% and 50.55% on the full dataset, both of which surpass Gemini-2.0-Flash. This underscores the pivotal role of large-scale, high-quality data in fine-grained emotion recognition.

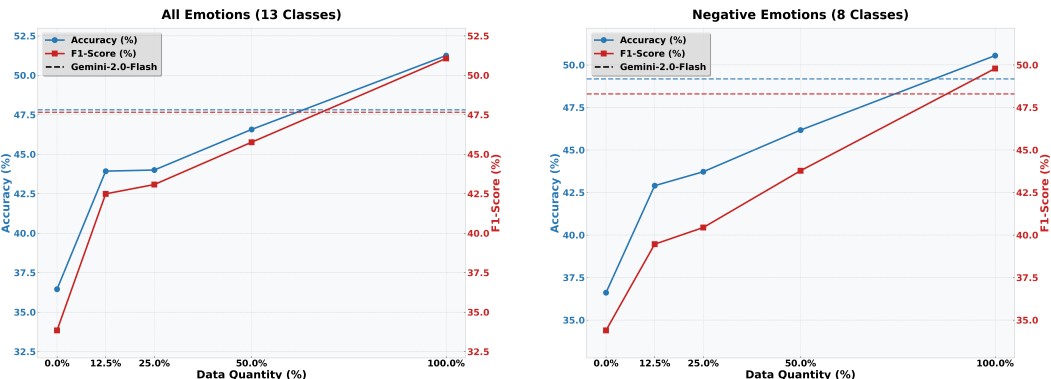

Figure 4: Impact of training data size on Libra-Emo Bench performance: left for all emotions, right for negative emotions. Dashed lines show Gemini-2.0-Flash results. Blue: Accuracy, Red: F1-Score.

## 4.3  FAILURE CASE ANALYSIS

As shown in Figure 5, due to small inter-class differences, our model performs poorly in distinguishing between *despairful* and *sad*, as well as *hateful* and *angry*. However, effectively differentiating these fine-grained emotions is crucial, as they represent distinct psychological states and behavioral tendencies: the former are deeper and more enduring, often accompanied by negative expectations or aggressive intent, while the latter are transient emotional fluctuations. Accurately distinguishing these emotions enables models to better understand human affect, supporting more targeted interventions and guidance in applications such as mental health monitoring and social media analysis.

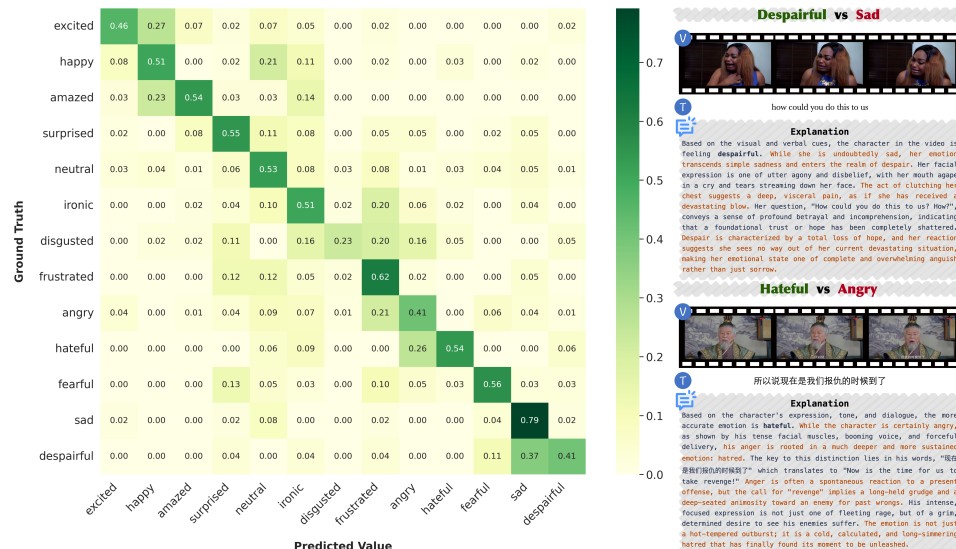

Figure 5: Failure case analysis of Libra-Emo-8B. Left: confusion matrix, Right: specific examples.

# 5 RELATED WORKS

**Multimodal Emotion Recognition Datasets.** Datasets such as MELD (Poria et al., 2019a), CMU-MOSEI (Zadeh et al., 2018a), and IEMOCAP (Busso et al., 2008) have advanced research but are limited in scale and granularity. MELD contains 13K utterances with 7 emotion categories; CMU-MOSEI has 23.5K clips with 6 basic emotions; IEMOCAP includes 10K samples across 9 categories. Recent benchmarks like EmoBench (Sabour et al., 2024) and EmotionQueen (Xu et al., 2024) highlight challenges in deep emotional understanding, while SemEval-2024 tasks (Saha et al., 2024) show negative emotions require more nuanced detection. Most datasets and models lack specialized training for fine-grained negative emotion recognition. In contrast, our work evaluates and fine-tunes MLLMs on Libra-Emo, improving recognition of subtle negative emotions.

**Multimodal Large Models for Emotion Recognition.** Recent multimodal large models such as CLIP (Radford et al., 2021) and its emotion-specific variants (EmoCLIP (Jiang et al., 2023), Emotion-LLaMA (Cheng et al., 2024)) show promise by integrating visual and textual data. Advanced vision-language models like PaLI (Chen et al., 2023a), Flamingo (Alayrac et al., 2022), and BLIP (Li et al., 2022) improve these capabilities. However, LLMs struggle with implicit emotional cues, especially negative emotions, exhibiting higher confusion among negative categories (Sabour et al., 2024; Saha et al., 2024). We address these issues by fine-tuning MLLMs such as InternVL (Chen et al., 2023b) on Libra-Emo Trainset, significantly enhancing fine-grained negative emotion recognition.

**Fine-grained Emotion Analysis.** Recent works highlight the need for fine-grained emotion analysis. GoEmotions (Demszky et al., 2020) offers a text-only dataset with 28 categories, and Emotic (Kosti et al., 2017) provides 26 emotion categories for images. However, these are limited to single modalities and often underrepresent negative emotions. Benchmarks like EmoBench and EmotionQueen (Sabour et al., 2024; Xu et al., 2024) extend evaluation to emotional intelligence in LLMs, stressing detection of implicit cues and subtle negative emotion distinctions. Our work advances this by focusing on fine-grained negative emotions in a multimodal setting, enabling nuanced video emotion analysis.

# 6 LIMITATIONS AND FUTURE WORK

Libra-Emo has several limitations: (1) suboptimal modality fusion methods; (2) limited evaluation in real-world applications such as content moderation and mental health assessment; and (3) high computational requirements. Despite these constraints, Libra-Emo provides a solid foundation for advancing negative emotion recognition research and facilitates more nuanced understanding of emotions in multimodal contexts. Future work will focus on addressing these limitations to further enhance the performance and applicability of fine-grained emotion recognition.

# 7 ETHICS STATEMENT

Our video data comes from YouTube and undergoes strict screening to ensure compliance with Creative Commons licenses. We will include video metadata in the released dataset to guarantee proper copyright attribution. Additionally, our models have potential risks, including privacy and data security concerns, misclassification of emotions and biases, as well as ethical and fairness challenges. Therefore, taking rigorous measures in data privacy protection, diverse modeling, and ethical review is crucial for ensuring the safe and fair application of this technology.

# 8 REPRODUCIBILITY STATEMENT

We commit to open-sourcing the Libra-Emo dataset, models, and code, and we have documented the implementation details thoroughly in the appendix.

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

## A  DATA SOURCE STATISTICS

Table 9: Summary of video data source statistics

| Category | # Videos | Category | # Videos |
|---|---|---|---|
| *Theme* | | | |
| Drama | 59 | Horror | 20 |
| Comedy | 55 | Sci-Fi | 18 |
| Action | 50 | Adventure | 17 |
| Mystery | 33 | Family | 10 |
| Romance | 32 | Fantasy | 9 |
| War | 32 | School | 5 |
| Crime | 24 | | |
| Costume | 21 | | |
| *Language* | | | |
| English | 274 | Chinese | 111 |
| **Total** | | | **385** |

## B  DETAILED PROMPTS

**Prompt for Annotation**

> **Prompt for Annotation**
>
> **System Prompt**
> I want you to act as a video emotion annotator. Please accurately understand the video content and output the answer according to the prompt format. Do not output any other content.
>
> **User Prompt**
> <video>
> subtitle: {subtitle}
> The above are a few evenly sampled images from a video and the subtitles for the video, which may be the words spoken by the people in the video.
> Please accurately identify the emotional label expressed by the people in the video and provide an explanation. Emotional labels should be limited to: happy, excited, angry, disgusted, hateful, surprised, amazed, frustrated, sad, fearful, despairful, ironic, neutral. This explanation should be accurate and concise.
> The output format should be: [label];[explanation]. Please do not output any additional content.

**Prompt for Explanation Synthesis**

> **Prompt for Explanation Synthesis**
>
> **System Prompt**
> I want you to act as a video emotion annotator. Please accurately understand the video content and output the answer according to the prompt format. Do not output any other content.
>
> **User Prompt**
> <video>
> subtitle: {subtitle}
> The above are a few evenly sampled images from a video and the subtitles for the video,

which may be the words spoken by the people in the video.
The emotion expressed by the person in the video is \*\*{emotion}\*\*. Please provide an explanation, describing the video content and the reasons for labeling it with this emotion. The output should be in JSON format:
{{
    "emotion": "{emotion}",
    "explanation": "Your answer"
}}

**Prompt for Fine-tuning**

> Prompt for Fine-tuning
>
> **User**
> <video>
> The above is a video. [if subtitle is not None] The video's subtitle is '{subtitle}', which maybe the words spoken by the person. [endif] Please accurately identify the emotional label expressed by the people in the video. Emotional labels include should be limited to: happy, excited, angry, disgusted, hateful, surprised, amazed, frustrated, sad, fearful, despairful, ironic, neutral. The output format should be:
> [label]
> [explanation]
>
> **Assistant**
> {label}
> {explanation}

## C  COMPLETE CONSTRUCTION DETAILS OF THE LIBRA-EMO TRAINSET

As shown in Table 10, the construction of the Libra-Emo Bench from the original pool of 64,824 candidate samples leaves 64,174 samples. In the 0th round of active learning, three models perform voting-based annotation combined with human annotation, and 2,549 samples are discarded due to quality issues, resulting in 61,625 samples. In the 1st round of active learning, 13,000 samples are selected for human annotation. Among them, 7,533 samples undergo label changes. In the 2nd round of active learning, 11,764 samples are selected for human annotation. Of these, 6,373 samples have their labels modified. After the explanation synthesis process, the Libra-Emo Trainset ultimately contains 61,625 meticulously processed samples.

Table 10: Complete construction details of the Libra-Emo Trainset.

| Operate | # Input | # Sample | # Drop | # Changed Labels | # Output |
|---------|---------|----------|--------|------------------|----------|
| Active Learning Round 0 | 64,174 | - | 2549 | - | 61,625 |
| Active Learning Round 1 | 61,625 | 13,000 | 0 | 7,533 | 61,625 |
| Active Learning Round 2 | 61,625 | 11,764 | 0 | 6,373 | 61,625 |
| Explanation Synthesis | 61,625 | - | 0 | - | **61,625** |

# D PROCESS OF THE HUMAN-MACHINE COLLABORATIVE ACTIVE LEARNING ANNOTATION STRATEGY

---

**Algorithm 1** Human-Machine Collaborative Active Learning Annotation Strategy

---

1: **Input:** Initial unlabeled dataset $D_{unlabeled}$, initial models $M_1, M_2, M_3$
2: **Output:** Labeled dataset $D_{labeled}$, trained model $M_{final}$
3: Initialize an empty labeled dataset: $D_{labeled} \leftarrow \emptyset$
4: **Step 1: Initial Labeling**
5: **for** each sample $x_i \in D_{unlabeled}$ **do**
6:     Get the predicted label from each model: $y_{i1}, y_{i2}, y_{i3} \leftarrow M_1(x_i), M_2(x_i), M_3(x_i)$
7:     Assign the initial label $y_i \leftarrow \text{vote}(y_{i1}, y_{i2}, y_{i3})$
8:     **if** the majority vote is successful (e.g., at least 2 models agree) **then**
9:         Add $(x_i, y_i)$ to $D_{labeled}$
10:     **else**
11:         Conduct human annotation on $x_i$ to obtain $y_i^{human}$
12:         Add $(x_i, y_i^{human})$ to $D_{labeled}$
13:     **end if**
14: **end for**
15: **Step 2: Iterative Label Refinement**
16: **repeat**
17:     **Step 2.1: Model Training**
18:     Train model $M_{current}$ using $D_{labeled}$
19:     **Step 2.2: Sample Selection**
20:     **for** each sample $(x_i, y_i) \in D_{labeled}$ **do**
21:         Predict new label $y_i^{current} \leftarrow M_{current}(x_i)$
22:         **if** $y_i^{current} \neq y_i$ **then**
23:             Add $(x_i, y_i)$ to $D_{new}$
24:         **end if**
25:     **end for**
26:     **Step 2.3: Human Annotation**
27:     **for** each sample $(x_i, y_i) \in D_{new}$ **do**
28:         Conduct human annotation on $x_i$ to obtain $y_i^{human}$
29:         Update $y_i$ with $y_i^{human}$
30:     **end for**
31:     Update the labeled dataset: $D_{labeled} \leftarrow D_{labeled} \setminus D_{new}$
32: **until** Model performance reaches saturation
33: **Step 3: Final Model**
34: Train final model $M_{final}$ using the fully labeled dataset $D_{labeled}$
35: **return** $M_{final}, D_{labeled}$

---

# E EXPERIMENTAL SETTING DETAILS

**Model Descriptions**

- **LLaVA-Video-7B-Qwen2**(Zhang et al., 2024a): Based on the Qwen2(qwe, 2024) as the foundation large language model, it supports a context length of 32K tokens and can process up to 64 video frames. It accepts joint inputs of videos, images, and multiple images.

- **Qwen2.5-VL-7B**(Team, 2025): Compared to Qwen2-VL(Wang et al., 2024), it incorporates dynamic frame rate (FPS) training and absolute temporal encoding techniques, enhancing the model's perception of temporal and spatial scales while further simplifying the network architecture to improve efficiency.

- **Phi-3.5-vision-instruct (4.2B)**(Abdin et al., 2024): With only 4.2B parameters, it concurrently processes text, images, and videos through attention mechanisms that align textual and visual modalities.

- **MiniCPM-o 2.6 (8B)**(Yao et al., 2024): Adopts an end-to-end omnimodal architecture capable of processing diverse inputs including text, images, audio, and video, while supporting real-time streaming interaction. Additionally, it offers multiple deployment options with low inference latency.

- **Qwen2.5-Omni-7B**(Xu et al., 2025): An end-to-end omnimodal model based on Qwen2.5, capable of processing text, images, audio, and video simultaneously. It uses TMRoPE positional encoding to align multimodal inputs and achieves strong performance on OmniBench, surpassing similar-scale single-modal models.

- **InternVL-2.5 series (1B-8B)**(Chen et al., 2024): Utilizes a Progressive Scaling Strategy for pretraining and extends dynamic high-resolution training methods to enhance capabilities in processing multi-image and video datasets.

**Fine-tuning Hyperparameters**

Table 11: Fine-tuning hyperparameters used in our experiments.

| Hyperparameter | Value |
|---|---|
| Learning Rate | 3e-4 |
| Learning Rate Schedule | Linear warmup + cosine decay |
| Warmup Ratio | 0.03 |
| Batch Size | 128 |
| Gradient Accumulation Steps | 1 |
| Training Epochs | 1 |
| Optimizer | AdamW |
| Weight Decay | 0.01 |
| Max Gradient Norm | 1.0 |
| GPU Type | H800 |
| GPU Memory | H800 |
| GPU Numbers | 32 |
| Trainging Time (8B) | 2h |

Table 12: Video Preprocessing Hyperparameters before Fine-tuning used in our experiments.

| Hyperparameter | Value |
|---|---|
| Frame sampling strategy | 16 frames uniformly distributed |
| Frame resolution | 448 × 448 pixels |
| Maximum subtitle length | 8192 tokens |
| Text tokenization | Model-specific tokenizer |
| Data augmentation | Random horizontal flip, color jitter |

# F  DETAILED CATEGORY DISTRIBUTION

Table 13: The detailed category distribution of the Libra-Emo dataset.

| Emotion Type | Emotion (7-CLS) | Emotion (13-CLS) | Quantity | | |
|---|---|---|---|---|---|
| | | | **Training** | **Testing** | **Total** |
| Positive | Happy | Excited | 819 | 41 | 860 |
| | | Happy | 5,119 | 61 | 5,180 |
| | | Amazed | 468 | 35 | 503 |
| Neutral | Surprised | Surprised | 2,194 | 62 | 2,256 |
| | Neutral | Neutral | 23,681 | 77 | 23,758 |
| Negative | Disgusted | Ironic | 5,348 | 51 | 5,399 |
| | | Disgusted | 420 | 44 | 464 |
| | Angry | Frustrated | 7,137 | 42 | 7,179 |
| | | Angry | 5,720 | 80 | 5,800 |
| | | Hateful | 525 | 35 | 560 |
| | Fearful | Fearful | 2,408 | 39 | 2,447 |
| | Sad | Sad | 7,152 | 48 | 7,200 |
| | | Despairful | 634 | 27 | 661 |
| **Total** | | | **61,625** | **642** | **62,267** |

# G  DEMONSTRATION OF THE ANNOTATION TOOL

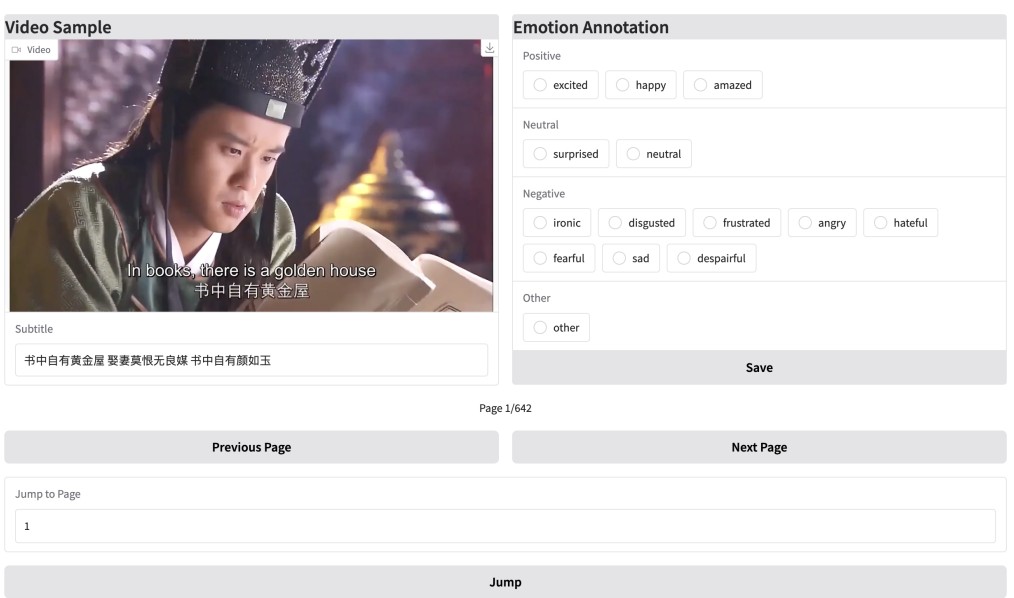

Figure 6: Annotation tool used in Libra-Emo.

## H MORE TRAINING EXAMPLES DEMONSTRATION

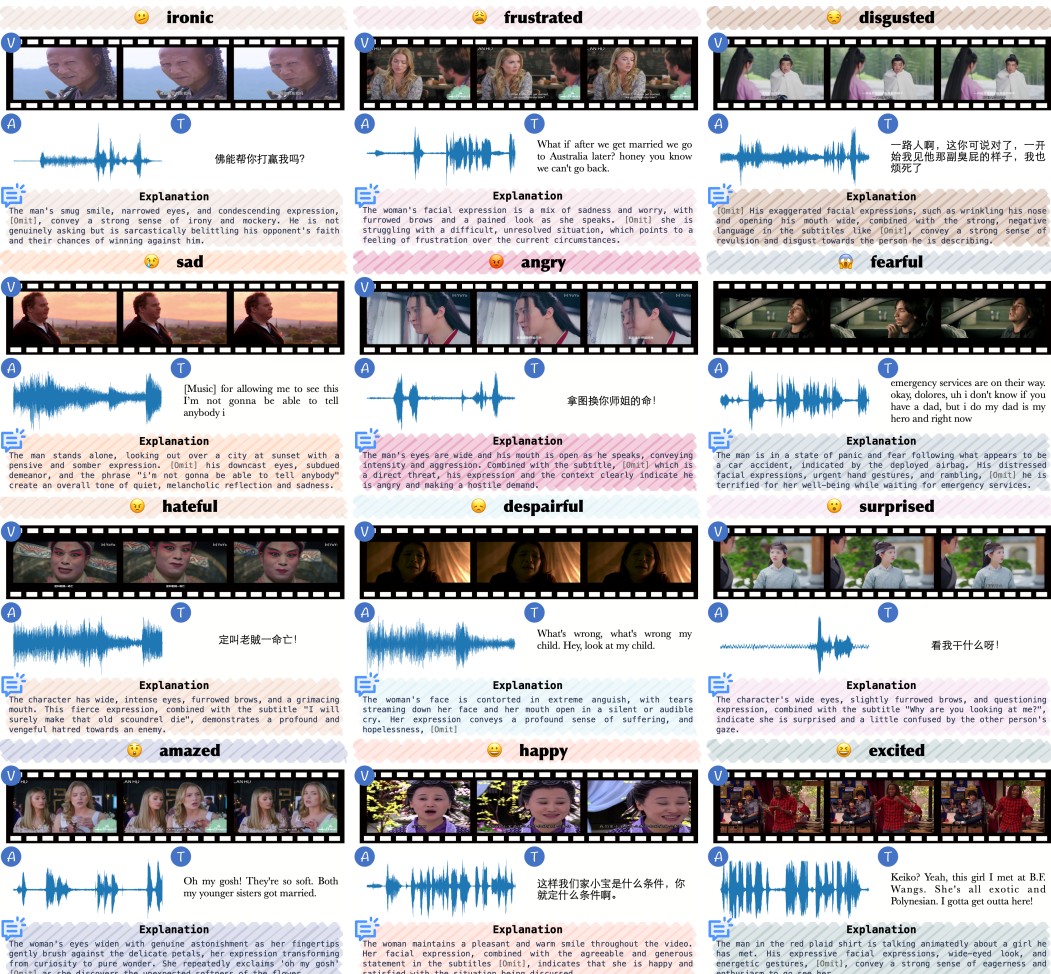

Figure 7: More training examples in Libra-Emo dataset.

## I USE OF LLMS

In our manuscript, we partially used large language models (LLMs) for academic polishing, but only to a limited extent.

