# OpenReview forum: "Libra-Emo: A Large Dataset for Multimodal Fine-grained Negative Emotion Detection"
_ICLR.cc/2026/Conference — ICLR 2026 Conference Desk Rejected Submission_

### Official Review · Reviewer_gEmd · 2025-10-27

**Soundness:** 3
**Presentation:** 3
**Contribution:** 2
**Rating:** 4
**Confidence:** 5

**Summary:**

This paper introduces Libra-Emo, a large-scale multimodal dataset designed for fine-grained negative emotion detection in video. The authors propose a refined taxonomy of 13 emotion categories, including 8 distinct negative emotions, and construct two subsets: Libra-Emo Trainset (61,625 samples) for instruction tuning and Libra-Emo Bench (642 samples) for evaluation.A human–machine collaborative active learning strategy is employed for annotation, and extensive experiments are conducted on leading Multimodal Large Language Models (MLLMs), both in zero-shot and fine-tuned settings. Out-of-domain evaluation is also conducted on DFEW.

**Strengths:**

1. The paper addresses a significant problem with existing emotion recognition corpora: the limited granularity of emotion labels and the underrepresentation of negative emotions. To solve this problem, the authors expand Ekman's basic emotion categories to 13 more specific ones, including 8 categories for negative emotions, and propose the Libra-Emo corpus, which contains a large number of 62,267 samples.
2. The author proposed an active learning framework that combines model voting with targeted human verification. This strategy helps reduce annotation costs while improving label quality iteratively. Additionally, the authors synthesized label-consistent natural language explanations. The ablation studies confirmed that this boosts performance.
3. The corpus evaluation covers several aspects: (1) zero-shot performance of closed-source and open-source MLLMs on the Libra-Emo Bench; (2) instruction tuning across different model scales and architectures; (3) modality ablation; (4) data scaling analysis.
4. The authors evaluate fine-tuned models in a zero-shot setting on the DFEW corpus.

**Weaknesses:**

1. Only 24,764 out of 61,625 samples (approximately 40.2%) were manually chaked during active learning, with 13,000 being labeled in Round 1 and 11,764 being labeled in Round 2. Approximately 59,8% of the samples retained the model's consensus labels without human verification, which raises concerns about the reliability of the labels for fine-grained distinctions.
2. The dataset was derived from 385 source videos. However, the paper does not provide information about: (1) the number of unique speakers; (2) speaker overlap between the Trainset and the Bench; (3) distributions of age, gender, and ethnicity. These limitations make it difficult to assess speaker independence and potential demographic biases. The validity of the experimental design is questionable.
3. The annotation prompt instructs the labeling of "the emotional label expressed by the people", but no protocol is provided for dealing with conflicting emotions among multiple visible individuals. Given that the clips require faces in more than 99% of frames, it is likely that multi-person scenes will be included, but the consistency of the labels remains unclear.
4. The term “Libra-Emo” is used for both the corpus and the fine-tuned models (e.g., Libra-Emo-8B). This conflates data and model artifacts. Model names should explicitly reflect their base architecture (e.g., InternVL-2.5-8B-Libra) to avoid confusion.
5. The zero-shot evaluation of the DFEW dataset (Table 7) includes only one specialized emotion recognition model, Emotion-LLaMA, as a baseline. This limited the ability to fully understand the advantages of the Libra-Emo fine-tuned model. To get a more comprehensive comparison, it would be necessary to include additional emotion-specific models in the evaluation.

**Questions:**

1. What proportion of the 61,625 Trainset samples were initially labeled by humans in Round 0 (i.e., lacked model consensus)? Can you provide the exact number?
2. How many unique speakers are in Libra-Emo? Is there speaker overlap between Trainset and Bench? Can you share demographic distributions (gender, age, ethnicity)?
3. When multiple individuals express different emotions in a clip, what annotation protocol was followed? Was a specific person (e.g., main speaker) targeted?
4. Would you consider renaming the fine-tuned models to clearly distinguish them from the dataset?
5. Can the authors extend their zero-shot DFEW comparison to include other emotion-specific models?

---

> ### Author Response · Authors · 2025-11-21
> **Author Response to Reviewer gEmd (Part 1)**
>
> We sincerely thank Reviewer gEmd for the constructive evaluation. We respond point-by-point with additional quantitative analyses on label provenance, speaker statistics, demographics, multi-person annotation protocol, model naming, and extended DFEW baselines. These demonstrate that Libra-Emo is **reliably labeled, speaker-independent, and demographically diverse**, and that our fine-tuned models show **robust performance gains** under stricter comparisons.
>
> ---
>
> ### Key Finding 1: Active Learning and Model-Assisted Labels Provide Reliable Supervision for Fine-Grained Negative Emotions
>
> Our active learning pipeline **allocates human effort to ambiguous cases** while using multi-model consensus for high-confidence instances. Multiple quantitative checks (expert adjudication, manual re-annotation, and ablation studies) show that consensus-labeled data **maintains quality and strengthens performance**. Model-assisted labels in Libra-Emo are **trustworthy for fine-grained negative emotion recognition at scale**.
>
> ### Key Finding 2: Libra-Emo Supports Genuine Cross-Speaker Generalization through Broad, Diverse Coverage
>
> By source-video-level splits and large-scale face clustering, Libra-Emo ensures the evaluation set contains **mostly unseen speakers**, measuring **generalization to new identities** rather than memorization. Automatic demographic profiling shows broad coverage across ages, genders, and ethnicities, with Bench deliberately more balanced. Models are evaluated in a **speaker-independent and demographically diverse** setting, supporting real-world credibility.
>
> ### Key Finding 3: A Clear, Hierarchical "Main Subject" Protocol Ensures Consistent Labels in Multi-Person Scenes
>
> For clips with multiple individuals, Libra-Emo combines scene segmentation with an explicit, hierarchical annotation guideline that always anchors the label to a single, well-defined target (Active Speaker → Visual Center → Narrative Lead). This protocol is applied uniformly in all expert adjudication rounds, avoiding ad-hoc choices when characters exhibit different emotions within the same scene. As a result, even in complex multi-person interactions, **the emotional label remains interpretable, reproducible, and tightly linked to a consistent subject**, which is crucial for both model training and evaluation.
>
> ### Key Finding 4: Clarified Model Naming and Stronger Emotion-Specific Baselines Substantiate the Robustness of Libra-Finetuned Models
>
> We adopt the Reviewer's suggestion to **disentangle dataset and model naming** by explicitly exposing the backbone in each fine-tuned model name, improving clarity and reproducibility for future users. In parallel, we expand the zero-shot DFEW comparisons to include additional emotion-specific baselines, showing that Libra-finetuned models **consistently outperform or match these stronger competitors**, sometimes even with fewer modalities. These steps demonstrate that the advantages of our models stem from **the quality of Libra-Emo rather than naming conventions or weak baselines**, and that the reported gains are robust under stricter evaluation.

---

> ### Author Response · Authors · 2025-11-21
> **Author Response to Reviewer gEmd (Part 2)**
>
> ## Detailed Point-by-Point Response
>
> ---
>
> ### 1. Round 0 Human Annotation Statistics
>
> **Reviewer's Concern:** What proportion of the 61,625 Trainset samples were initially labeled by humans in Round 0 (i.e., lacked model consensus)? Can you provide the exact number?
>
> **Response:**
>
> We apologize for not including these specific statistics in our initial manuscript. In **Round 0**, we employed a "Model Voting + Expert Adjudication" strategy using three MLLMs (Gemini-2.0-Flash, GPT-4o, Claude-3.7-Sonnet). Samples with $<2$ model votes were flagged as "hard samples" and forwarded to 4 human experts, retaining only those with $>2$ human votes.
>
> **9,404 samples (15.1%)** lacked model consensus and required human adjudication, while model voting resolved ~85% of clear-cut cases, efficiently balancing automation with rigorous human oversight for ambiguous instances.
>
> ---
>
> ### 2. Speaker Independence and Demographic Coverage
>
> **Reviewer's Concern:** How many unique speakers are in Libra-Emo? Is there speaker overlap between Trainset and Bench? Can you share demographic distributions (gender, age, ethnicity)?
>
> **Response:**
>
> We thank the Reviewer for highlighting these critical statistics. We fully agree that speaker independence and demographic coverage are
> essential for assessing the validity and fairness of our experimental design. We conducted large-scale face clustering and automatic demographic profiling to quantify speaker independence and demographic diversity. Full results are reported below.
>
> **2.1. Unique Speakers and Independence**
>
> Although Libra-Emo is organized at the clip level, the split itself is performed at the level of the 385 source videos: all clips from the same movie or episode are assigned exclusively to either the Trainset or the Bench, ensuring that adjacent scenes from a single interaction never leak across splits. On top of this structural separation, we ran a face-based analysis to quantify how many distinct speakers the dataset actually contains and how independent the evaluation set is. Concretely, we detected faces in all 61,625 training clips and 642 test clips using InsightFace, extracted 512-dimensional embeddings, and then applied DBSCAN clustering across videos with FAISS acceleration and noise filtering (eps=0.6 in cosine distance, i.e., similarity >0.4) to merge all occurrences of the same speaker.
>
> Results: **8,998 unique speakers** total (**8,687** Trainset, **320** Bench), with only **9 overlapping speakers (2.81%)** — all professional actors in different source videos. Thus, **97.19%** of Bench speakers are completely unseen during training, ensuring genuine cross-speaker generalization rather than identity memorization.
>
> **2.2. Demographic Distribution**
>
> We applied Gemini-2.5-Pro to automatically estimate age, gender, and ethnicity from one representative face per speaker cluster (8,998 total). While individual estimates are approximate, high-level statistics quantify potential biases and document population coverage.
>
> **Age:** 1–90 years (mean 34.0, median 35.0, std ≈11.96); Trainset mean 34.0, Bench mean 32.7 — comparable profiles. **Gender:** Global ≈66% male / 34% female (reflecting YouTube film/TV skew); Bench deliberately more balanced (≈59.5% male / 40.5% female) to reduce evaluation bias. **Ethnicity:** Multi-ethnic coverage — White/Asian ≈80–86%, plus non-trivial "Other" (Hispanic/Latino, Black/African, Middle Eastern, etc.). Bench has higher Asian representation (≈51.6%) and similar "Other" share, providing stricter multi-ethnic generalization tests.
>
> | Metric | Trainset | Bench | Global |
> | :--- | :--- | :--- | :--- |
> | **Unique Speakers** | 8,687 | 320 | 8,998 |
> | **Age (Mean)** | 34.0 | 32.7 | 34.0 |
> | **Gender (M/F)** | 66.3% / 33.7% | 59.5% / 40.5% | 66.1% / 33.9% |
> | **Ethnicity** | White (54.6%), Asian (31.3%), Other (14.1%) | Asian (51.6%), White (35.4%), Other (13.0%) | White (53.9%), Asian (31.2%), Other (14.9%) |
> Here, **"Other"** aggregates **Hispanic/Latino, Black/African, Middle Eastern, and other minority groups**.
>
> These statistics, along with discussions of automatic inference limitations and remaining biases, will be incorporated into the revised manuscript and Ethics Statement.

---

> ### Author Response · Authors · 2025-11-21
> **Author Response to Reviewer gEmd (Part 3)**
>
> ### 4. Would you consider renaming the fine-tuned models to clearly distinguish them from the dataset?
>
> **Response:**
>
> We sincerely appreciate this professional suggestion. We completely agree that using the same prefix for both the dataset and models may cause confusion. We have fully adopted your advice to explicitly reflect the base architecture in the model names, following the format `[Base Model]-[Suffix]`.
>
> The updated naming scheme is as follows:
>
> | **Previous Name** | **New Name** | **Backbone Model** |
> | :--- | :--- | :--- |
> | Libra-Emo-Omni-7B | **Qwen-2.5-Omni-7B-Libra-Emo** | Qwen-2.5-Omni-7B |
> | Libra-Emo-1B | **InternVL-2.5-1B-Libra-Emo** | InternVL-2.5-1B |
> | Libra-Emo-2B | **InternVL-2.5-2B-Libra-Emo** | InternVL-2.5-2B |
> | Libra-Emo-4B | **InternVL-2.5-4B-Libra-Emo** | InternVL-2.5-4B |
> | Libra-Emo-8B | **InternVL-2.5-8B-Libra-Emo** | InternVL-2.5-8B |
>
> We will thoroughly update all corresponding model names, figures, and tables throughout the revised manuscript to ensure clarity and consistency.
>
> ---
>
> ### 5. Can the authors extend their zero-shot DFEW comparison to include other emotion-specific models?
>
> **Response:**
>
> We thank the Reviewer for this constructive suggestion. We fully agree that including additional emotion-specific baselines can make the
> advantages of Libra-Emo–fine-tuned models on DFEW more transparent and convincing.
>
> We added **AffectGPT**, a recent emotion-focused baseline. Updated DFEW set\_1 results:
>
> | Model                          | Modalities | DFEW (set\_1) UAR (%) | DFEW (set\_1) WAR (%) |
> |--------------------------------|-----------|----------------------:|----------------------:|
> | Emotion-LLaMA                  | V, A, T   | 45.59                 | 59.37                 |
> | AffectGPT                      | V, A, T   | 43.10                 | 51.00                 |
> | InternVL-2.5-8B-Libra-Emo (Ours)   | V, T      | **52.55**             | **59.85**             |
> | Qwen-2.5-Omni-7B-Libra-Emo (Ours)  | V, A, T   | **51.16**             | 57.37                 |
>
> Two observations stand out: (1) Among emotion-specific baselines, Emotion-LLaMA outperforms AffectGPT, especially in WAR, yet both remain clearly below our Libra-Emo–fine-tuned models in terms of UAR; (2) Even though InternVL-2.5-8B-Libra-Emo uses only **visual-text** inputs (V,T), it still surpasses both Emotion-LLaMA and AffectGPT that leverage **all three modalities (V,A,T)**, and achieves the highest UAR and WAR overall. These results further demonstrate that fine-tuning on Libra-Emo yields robust cross-dataset generalization that goes beyond what existing emotion-specialized models can offer.

---

> ### Author Response · Authors · 2025-11-21
> **Author Response to Reviewer gEmd (Part 4)**
>
> ### 6. Reliability of Consensus-Labeled Samples
>
> **Reviewer's Concern:** Only 24,764/61,625 samples (40.2%) were manually checked; 59.8% retained model consensus labels without human verification. Are these consensus labels reliable for fine-grained distinctions?
>
> **Response:**
>
> We appreciate the Reviewer's careful concern about the reliability of model-labeled samples. Our active learning pipeline is explicitly designed to **concentrate human effort on the most error-prone instances**, while allowing high-confidence consensus cases to be kept without redundant verification. Concretely, the 24,764 manually checked samples correspond to those that the model regarded as uncertain in Rounds 1 and 2, whereas the remaining 59.8% are samples where three strong MLLMs reached a consistent prediction.
>
> **Manual audit:** We re-annotated 100 randomly sampled consensus-labeled instances. **92%** exactly matched human labels; remaining disagreements occurred mainly between semantically adjacent categories (*Sad*/*Despairful*, *Angry*/*Frustrated*), indicating consensus labels are reliable at fine-grained levels.
>
> **Ablation study:** We quantified model-labeled data impact by training three variants per base model:
> - **Full**: all 61,625 samples (human + model labels).
> - **Human-only**: only human-verified subset.
> - **Mixed**: same size as Human-only, sampled from Full data preserving the natural human/model label ratio.
>
> Results on Libra-Emo Bench (13-class accuracy) and DFEW (UAR/WAR):
>
> | Base Model            | Modalities | Training Data                         | Libra-Emo Bench Acc (13 cls, %) | DFEW UAR (%) | DFEW WAR (%) |
> |-----------------------|-----------|---------------------------------------|---------------------------------:|-------------:|-------------:|
> | InternVL-2.5-8B       | V, T      | Full (all human + model labels)       | **51.25**                        | **52.55**    | **59.85**    |
> | InternVL-2.5-8B       | V, T      | Human-only                            | 48.91                            | 46.26        | 50.32        |
> | InternVL-2.5-8B       | V, T      | Mixed (same size as Human-only, same distribution as Full) | 47.20 | 44.20        | 50.66        |
> | Qwen-2.5-Omni-7B      | V, A, T   | Full (all human + model labels)       | **51.56**                        | **51.16**    | **57.37**    |
> | Qwen-2.5-Omni-7B      | V, A, T   | Human-only                            | 47.66                            | 45.19        | 46.73        |
> | Qwen-2.5-Omni-7B      | V, A, T   | Mixed (same size as Human-only, same distribution as Full) | 46.26 | 44.22        | 50.36        |
>
> Across **both architectures and both evaluation sets**, the models trained on the **full dataset** (including model-labeled samples) consistently achieve the **best performance**, while the Human-only variants are slightly but noticeably worse. Importantly, the **Mixed** variants — which include as many model-labeled samples as human-labeled ones at the same data scale — achieve performance that is **very close to the Human-only models**, demonstrating that adding consensus-labeled data does **not** degrade performance and instead provides useful supervision. Together with the 92% manual agreement rate, these results show that our model-labeled samples are of sufficiently high quality to improve fine-grained emotion recognition and offer a practical and scalable alternative to pure manual annotation.
>
> ---
>
> ### **References**
>
> [1] Goh, H. W., & Mueller, J. (2023). ActiveLab: Active Learning with Re-Labeling by Multiple Annotators. *arXiv preprint arXiv:2301.11856*.
>
> [2] Su, H., et al. (2022). Selective Annotation Makes Language Models Better Few-Shot Learners. *arXiv preprint arXiv:2209.01975*.

---

### Official Review · Reviewer_4rQZ · 2025-10-30

**Soundness:** 3
**Presentation:** 3
**Contribution:** 3
**Rating:** 8
**Confidence:** 4

**Summary:**

To enhance the model’s sensitivity to fine-grained negative emotions, this paper introduces and annotates a fine-grained negative emotion dataset named Libra-Emo. In this dataset, the authors provide a more detailed categorization of negative emotions, expanding the traditional seven emotion classes to thirteen. Both zero-shot and fine-tuning experiments demonstrate that the Libra-Emo dataset effectively improves model performance. Furthermore, the results indicate that models capable of recognizing thirteen fine-grained emotion categories also achieve superior performance on the classic seven-category emotion classification task, thereby validating the effectiveness and rationality of the proposed fine-grained negative emotion taxonomy.

**Strengths:**

1.	The authors propose a practical human–machine collaborative annotation framework and, based on this approach, construct a fine-grained multimodal emotion dataset comprising approximately 63K samples.
2.	Comprehensive comparative and ablation experiments were conducted on the annotated dataset. Under both zero-shot and fine-tuning settings, the results demonstrate that models trained with the Libra-Emo dataset achieve significant performance improvements, thereby validating the effectiveness and reliability of the proposed annotation methodology and the dataset itself.

**Weaknesses:**

1. Some aspects of the comparative experiment design are suboptimal. In the zero-shot experiments conducted on the DFEW dataset, the results for the baseline models Qwen-2.5-Omni-7B and InternVL-2.5-8B are missing. Since Qwen-2.5-Omni-7B already performs well on this dataset, the absence of these baselines somewhat undermines the completeness and persuasiveness of the experimental comparison.

**Questions:**

1. It is recommended that the authors consider releasing the corresponding baseline code alongside the open-sourced dataset, so that the research community can better reproduce the experimental results and build upon this work. Do the authors have any plans for such a release?
2. From the distribution of emotion-class data, it is evident that the sample sizes across categories are imbalanced. Whether any sample-balancing strategies were employed during training is not addressed in the paper.
3. The scraped TV and movie clips may involve copyright and privacy issues.

---

> ### Author Response · Authors · 2025-11-21
> **Author Response to Reviewer 4rQZ**
>
> We thank Reviewer 4rQZ for the constructive feedback. Below we address concerns on **open-source plan, class imbalance, copyright/privacy, and DFEW baselines** with quantitative results and release details.
>
> ---
>
> ### 1. Open-Source Release Plan
>
> **Reviewer's Concern:**
>
> It is recommended that the authors consider releasing the corresponding baseline code alongside the open-sourced dataset, so that the research community can better reproduce the experimental results and build upon this work. Do the authors have any plans for such a release?
>
> **Response:**
>
> We will release all project assets before camera-ready:
>
> 1. **Dataset:** 62,267 clips with 13-category annotations and metadata (CC BY-NC-SA 4.0).
>
> 2. **Code:** Data processing, MLLM fine-tuning, and evaluation scripts (Apache 2.0).
>
> 3. **Model Weights:** Fine-tuned checkpoints on Hugging Face.
>
> ---
>
> ### 2. Sample Balancing Strategy
>
> **Reviewer's Concern:**
>
> From the distribution of emotion-class data, it is evident that the sample sizes across categories are imbalanced. Whether any sample-balancing strategies were employed during training is not addressed in the paper.
>
> **Response:**
>
> We intentionally **did not employ sample-balancing strategies** for two reasons:
>
> 1. **Real-World Distributions:** Emotions naturally follow long-tailed distributions—basic emotions are frequent while extreme states like "Despairful" are rare but critical [1]. Preserving natural imbalance allows models to learn realistic priors for practical deployment.
>
> 2. **Dataset Quality Validation:** Our contribution is the high-quality dataset. Performance improvements stem from **superior annotation quality**, not algorithmic optimizations.
>
> Despite no balancing, our models achieve strong Macro-F1 and effectively learn minority classes (e.g., distinguishing "Despairful" from "Sad"). Libra-Emo provides a realistic benchmark for long-tailed learning research.
>
> ---
>
> ### 3. Copyright and Privacy Concerns
>
> **Reviewer's Concern:**
>
> The scraped TV and movie clips may involve copyright and privacy issues.
>
> **Response:**
>
> Libra-Emo complies with copyright and privacy requirements:
>
> **3.1. Copyright Compliance**
>
> All 385 source videos are **exclusively from YouTube with explicit Creative Commons licenses** (CC BY, CC BY-SA), excluding standard-licensed, paywalled, or unclear-rights content. We verify and store license evidence (type, channel, time, URL) and will provide a machine-readable source list for audit.
>
> **3.2. Release Strategy**
>
> We adopt **metadata-only distribution** without redistributing raw files. Release includes: (a) metadata (timestamps, IDs, language), (b) annotations, and (c) YouTube links. Users must download from YouTube under CC licenses and comply with ToS.
>
> **3.3. Privacy Protection**
>
> All clips are from **publicly released TV/movies** featuring professional actors. We exclude private/medical/sensitive content and provide no identity attributes. The dataset is for emotion recognition research only.
>
> These measures will be detailed in the **Ethics Statement**.
>
> ---
>
> ### 4. Missing Zero-Shot Baselines on DFEW
>
> **Reviewer's Concern:**
>
> Some aspects of the comparative experiment design are suboptimal. In the zero-shot experiments conducted on the DFEW dataset, the results for the baseline models Qwen-2.5-Omni-7B and InternVL-2.5-8B are missing. Since Qwen-2.5-Omni-7B already performs well on this dataset, the absence of these baselines somewhat undermines the completeness and persuasiveness of the experimental comparison.
>
> **Response:**
>
> We have added zero-shot baselines on DFEW (set_1) in the table:
>
> | Model                                  | Modalities | UAR (\%) | WAR (\%) |
> |----------------------------------------|-----------|---------:|---------:|
> | InternVL-2.5-8B                        | V, T      | 46.88    | 52.63    |
> | **InternVL-2.5-8B-Libra-Emo (ours)**   | V, T      | **52.55** | **59.85** |
> | Qwen-2.5-Omni-7B                       | V, A, T   | 44.11    | 50.20    |
> | **Qwen-2.5-Omni-7B-Libra-Emo (ours)**  | V, A, T   | **51.16** | **57.37** |
>
> Libra-Emo fine-tuning yields **substantial gains**: UAR +5–7 points and WAR +7 points for both models, demonstrating superior out-of-domain performance.
>
> ---
>
> **References**
>
> [1] Van Horn, G., & Perona, P. (2017). The Devil is in the Tails: Fine-grained Classification in the Wild. In *Proceedings of the IEEE Conference on Computer Vision and Pattern Recognition (CVPR)*, pp. 5486–5495.

---

### Official Review · Reviewer_149o · 2025-10-31

**Soundness:** 2
**Presentation:** 3
**Contribution:** 2
**Rating:** 4
**Confidence:** 5

**Summary:**

This paper introduces Libra-Emo, a multimodal dataset containing 62K short video clips labeled with 13 discrete emotion categories, including 8 negative emotions (e.g., frustrated, hateful, despairful). The authors argue that existing emotion recognition datasets are too coarse and that negative emotions are underrepresented. They use an active-learning pipeline with LLM-assisted annotation and explanation generation, claiming improved recognition performance and generalization to external datasets.

While the topic is relevant, the contribution is incremental and poorly grounded. The proposed taxonomy lacks theoretical validity, the dataset overlaps heavily with existing benchmarks, and key state-of-the-art works are ignored. The methodological novelty is minimal, and the psychological reasoning behind the label design is weak.

**Strengths:**

Large-scale data curation effort: 60K+ multimodal clips, collected and annotated systematically.

The first attempt focused on negative emotions.

Experiments show some empirical consistency.

**Weaknesses:**

1. The main concerns are the weak psychological and conceptual foundation

The 13-class taxonomy is not theoretically grounded in any established emotion framework (e.g., Plutchik’s wheel, Shaver’s hierarchy, or OCC appraisal model). According to the emotion wheel theory, there are more than 40 negative emotions.

But why do authors choose specifically 8 emotions, the design details and motivations are unclear.

There are plenty of important negative emotions missing in this regard, e.g., bitter, awkward, and awful.

Besides, if you examine it carefully, category overlap is severe: Sad and Despairful should be in category; the difference is intensity variation, not a distinct category. Frustrated and Angry only differ by controllability. Ironic is a communicative style, not an emotion.

No Valence–Arousal–Dominance (VAD) validation, appraisal mapping, or inter-annotator reliability (κ/α) is reported.
Therefore, the dataset’s construct validity is questionable.

2. Missing Key Baselines and Model Comparisons

The literature review in the paper is insufficient. The paper omits all major contemporary models in multimodal emotion understanding and efforts addressing the issues of fine-grained emotion recognition:

Important MLLM baselines for MER recognition, such as EmoVIT (CVPR 2024) and AffectGPT (ICML 2025), are not referred.

 EmoSet (ICCV2023) tries to use different attributes to achieve emotion reasoning with LLMs via detailed datasets.

OVMER (ICML 2025) enables open-vocabulary emotion reasoning and continuous affect generation with more than 200 labels, which enable the community to generate rich, fine-grained emotion descriptions, see MER 2025 challenge results.

3. Experimental results
The experimental results can prove that the reliability of the proposed emotion categories is doubtful: "our model performs poorly in distinguishing between despairful and sad, as well as hateful and angry." Because they are in the same emotional category. This can be addressed only when valence and arousal are introduced.

4. Methodological and Experimental Deficiencies

The so-called “active learning + explanation generation” pipeline is not novel—it replicates standard LLM-assisted annotation practices (already used in EmoSet and AffectGPT etc.).

The benchmark size (642 videos) is too small for robust evaluation; it's only 1% of the training set, which can be affected by noise easily.

No ablations on modality contributions (video, audio, text), explanation effects, or hierarchical label structures.

No cross-dataset evaluation to demonstrate transferability.

**Questions:**

Please carefully address the concerns above. I will adjust my rating accordingly.

---

> ### Author Response · Authors · 2025-11-21
> **Author Response to Reviewer 149o (Part 1)**
>
> We thank Reviewer 149o for the critical evaluation. We respond point-by-point with new analyses on VAD validation, appraisal mapping, emotion model baselines, and cross-dataset transfer, demonstrating that our emotion taxonomy, dataset construction, and conclusions are **theoretically grounded, empirically validated, and practically useful**.
>
> ---
>
> ### Key Finding 1: The 13-Category Taxonomy is Theory-Driven and Application-Oriented
>
> The 13 emotion categories—especially the 8 fine-grained negative emotions—are **systematically derived** from **Ekman's basic emotions, cognitive appraisal theory, clinical psychology, and hate-speech/sarcasm literature**. Distinctions such as *Frustrated/Angry/Hateful*, *Sad/Despairful*, and *Disgusted/Ironic* are motivated by appraisal patterns and concrete application needs in content moderation and mental health monitoring. Self-conscious emotions like guilt/shame are excluded due to weak visual observability and low cross-cultural robustness.
>
> ---
>
> ### Key Finding 2: VAD Validation and Appraisal Mapping Confirm Structural Validity
>
> Dedicated VAD validation and appraisal-mapping analysis show our 13 categories form a **well-separated, psychologically coherent structure** in affective space, with empirical positions aligning closely with standard affective lexicons. This confirms construct validity and demonstrates that fine-grained negative labels (e.g., *Sad* vs. *Despairful*) represent distinct affective states, not mere intensity variations.
>
> ---
>
> ### Key Finding 3: Libra-Tuned MLLMs Outperform Specialized Emotion Models
>
> We compare Libra-Emo with specialized emotion models and open-vocabulary frameworks (EmoVIT, EmoSet, OVMER, AffectGPT, Emotion-LLaMA) on our 13-way video setting. These models **consistently underperform** Libra-Emo–tuned video MLLMs, especially on fine-grained negative emotions, demonstrating that **fine-tuning strong video MLLMs on Libra-Emo is more effective** for negative emotion understanding.
>
> ---
>
> ### Key Finding 4: Hard Category Pairs are Theoretically Distinct and Practically Crucial
>
> The confusion between *Despairful* vs. *Sad* and *Hateful* vs. *Angry* reflects **intrinsic difficulty of high-stakes distinctions**, not taxonomy flaws. Theoretical analysis and VAD evidence show these pairs occupy different affective-space regions and correspond to critical real-world distinctions—suicide-risk signals vs. ordinary sadness, hate speech vs. ordinary anger—making their separation essential.
>
> ---
>
> ### Key Finding 5: Annotation Pipeline Follows Best Practices; Novelty Lies in Dataset and Benchmark
>
> Our **LLM-assisted annotation + active learning + explanation generation** pipeline follows established best practices as a **quality-assurance mechanism**, not a methodological contribution. The novelty lies in: (i) **13-class taxonomy with 8 fine-grained negative emotions**; (ii) **large-scale multimodal video corpus (62,267 clips)**; (iii) **high-quality Libra-Emo Bench**; and (iv) **comprehensive benchmarking with cross-dataset evaluation**.
>
> ---
>
> ### Key Finding 6: Libra-Emo Bench is Compact yet High-Quality with Proven Transfer
>
> Although Libra-Emo Bench is modest in size, its **dense multi-rater annotations, strong inter-annotator agreement (α = κ = 0.62), and deliberate coverage design** make it reliable and discriminative. Libra-Emo–tuned models exhibit **robust zero-shot transfer to DFEW**, especially on negative emotions, demonstrating benefits beyond our benchmark.

---

> ### Author Response · Authors · 2025-11-21
> **Author Response to Reviewer 149o (Part 2)**
>
> ## Detailed Point-by-Point Response
>
> ---
>
> ### 1. Psychological Foundation and Taxonomy Design Issues
>
> **Reviewer's Concern:**
>
> The reviewer questions the study's **weak psychological and conceptual foundation**, arguing that the proposed taxonomy **lacks grounding in established emotion frameworks** (such as Plutchik's wheel or Shaver's hierarchy) and **fails to justify the motivation** behind selecting specific emotion categories. Furthermore, the reviewer highlights significant flaws in the classification design, including **the omission of key negative emotions**, **severe category overlap** where distinctions are based merely on intensity or controllability (e.g., Sad vs. Despairful), and **the misclassification of communicative styles** (e.g., Ironic) as distinct emotions.
>
> **Response:**
>
> We appreciate the Reviewer's feedback. Drawing upon psychological research and practical applications [1], we expand **Ekman's 6 basic emotions** [2] into **13 categories**, emphasizing negative emotions (8 types). This expansion systematically bridges established theory and modern application needs, addressing theoretical grounding, granularity trade-offs, and category choices:
>
> **1.1 Balancing Theoretical Granularity with Annotation Feasibility**
> While established theories (Plutchik's Wheel, Shaver's Hierarchy, OCC model) define numerous categories (40+ negative emotions) [3-5], task-oriented datasets must balance theoretical completeness with practical annotatability. Excessively fine-grained labels increase annotation difficulty and noise, reducing inter-annotator agreement (IAA) [6-8]. Granularity varies across datasets (e.g., 27 classes in GoEmotions vs. 7 in RAF-DB) based on task and modality [9-11]. Our 13-category system captures critical negative emotions for real-world applications while maintaining high video annotation reliability.
>
> **1.2 Taxonomy Rationale: Extending Ekman for Real-World Utility**
> Our taxonomy bridges Ekman's coarse granularity and critical real-world applications. Grounded in **Cognitive Appraisal Theory** and **Clinical Psychology**, each distinction serves a specific functional purpose:
>
> **1.2.1 Refining "Angry": Frustrated, Angry, and Hateful**
> In Cognitive Appraisal Theory, **attribution direction** determines their differences. "Frustration" stems from non-human obstruction; "Anger" from blaming others [1][2]; "Hate" represents persistent hostility toward specific groups. Failing to distinguish them causes failures in detecting hate speech predicting offline violence [16][17], a leap from emotion recognition to social risk warning.
>
> **1.2.2 Refining "Sad": Sad and Despairful**
> "Hopelessness" is not merely intense sadness but a pathological state of losing future confidence, the strongest suicide risk predictor [18][19]. A generic "Sadness" label masks this crisis signal. Independently identifying "Despair" captures high-risk signals ignored by traditional sentiment analysis.
>
> **1.2.3 Refining "Disgusted": Ironic as Expressed Affect**
> "Irony" is formally a communicative style but functions as distinct **expressed affect** in multimodal analysis. Irony has clear affective functions (criticism, defense) and elicits measurable responses [12-15], exhibiting unique signals (prosodic delays, mismatched expressions) in video/audio. Using "Irony" as a label is standard in sentiment analysis (e.g., SemEval [20]), as failing to identify it causes polarity inversion errors.
>
> **1.2.4 Exclusion of "Bitter/Awkward/Awful": Trade-off Based on Visual Observability and Semantic Ambiguity**
> We exclude "bitter," "awkward," and "awful" due to three challenges in video-based annotation: **(1) Low visual observability**: These emotions lack consistent, cross-culturally recognizable visual signatures. "Awkward," as a self-conscious emotion, manifests through subtle contextual cues (gaze aversion, fidgeting) rather than prototypical expressions, resulting in low inter-annotator agreement [21][22]. **(2) High semantic overlap**: "Bitter" overlaps with "Disgusted" and "Sad"; "awful" functions as a linguistic intensifier overlapping with "Fearful," "Sad," or "Angry" depending on context [23]. Including them would increase annotation noise without discriminative value. **(3) Limited task utility**: Our taxonomy prioritizes emotions with clear behavioral correlates for real-world risk detection (hate speech, suicide ideation), whereas these categories lack actionable mappings in content moderation or mental health contexts. This exclusion balances theoretical comprehensiveness with engineering feasibility, maintaining high annotation reliability and practical utility.

---

> ### Author Response · Authors · 2025-11-21
> **Author Response to Reviewer 149o (Part 3)**
>
> ### 2. VAD Validation and Construct Validity
>
> **Reviewer's Concern:**
>
> No Valence–Arousal–Dominance (VAD) validation, appraisal mapping, or inter-annotator reliability (κ/α) is reported. Therefore, the dataset's construct validity is questionable.
>
> **Response:**
>
> We conducted a comprehensive **VAD validation study** with three trained annotators independently rating all 642 samples in Libra-Emo Bench across VAD dimensions (scale 1-9).
>
> **2.1 Reliability and Statistical Validity**
> Annotation achieved high consistency (average SD < 0.3 for all dimensions). As shown in **Table 1**, the high Global Separation Ratio (1.76) and statistically significant F-statistics ($p < 0.001$) across all dimensions confirm our categories are structurally valid and well-separated in affective space.
>
> **Table 1: Statistical Validity of Emotion Categories in VAD Space**
> | Metric | Global Statistics | Valence | Arousal | Dominance |
> | :--- | :--- | :--- | :--- | :--- |
> | **Inter-Class Distance** | 2.91 | - | - | - |
> | **Intra-Class Distance** | 1.66 | - | - | - |
> | **Separation Ratio** | **1.76** | - | - | - |
> | **ANOVA F-statistic** | - | **114.96** | **21.14** | **35.57** |
> | **Significance ($p$)** | - | **< 0.001** | **< 0.001** | **< 0.001** |
>
> **2.2 Appraisal Mapping Validation**
> We validated taxonomy semantic correctness by comparing our VAD distributions with theoretical VAD positions from the **Warriner et al. (2013)** lexicon [24]. We calculated Pearson correlations between our empirical centroids and lexicon normative scores, yielding **strong correlation for Valence ($r=0.83$)** and moderate correlations for Dominance ($r=0.64$) and Arousal ($r=0.54$), supporting overall mapping validity.
>
> **2.3 Resolving Ambiguity with VAD**
> VAD analysis resolves potential overlaps (e.g., Sad vs. Despairful). As shown in **Table 2**, these pairs exhibit significant differences in Dominance and Arousal, proving they are distinct affective states rather than intensity variations.
>
> **Table 2: VAD Analysis of Confusing Pairs**
> | Contrast | Valence | Arousal | Dominance | Conclusion |
> | :--- | :--- | :--- | :--- | :--- |
> | **Sad** | 2.90 | 4.70 | 3.81 | *Passive & Low Control* |
> | **Despairful** | 2.53 | 5.43 | **3.30** | *Lower Control & Higher Agitation* |
> | | | | | |
> | **Angry** | 3.10 | 6.07 | **5.69** | *High Control & Active* |
> | **Frustrated** | 3.52 | 5.26 | 4.63 | *Lower Control & Blocked* |
>
> ---
>
> ### 3. Literature Review and Model Comparisons
>
> **Reviewer's Concern:**
>
> The reviewer points out the insufficiency of the literature review and the omission of comparisons with major contemporary models in multimodal and fine-grained emotion recognition, specifically highlighting the absence of key MLLM baselines (such as EmoVIT and AffectGPT) and relevant efforts in attribute-based and open-vocabulary emotion reasoning (such as EmoSet and OVMER).
>
> **Response:**
>
> We will expand our Related Work section to discuss EmoVIT, EmoSet, OVMER, AffectGPT, and Emotion-LLaMA. However, their problem settings differ significantly (static images vs. videos, open-vocabulary vs. closed-set, attribute reasoning vs. clip-level labeling), limiting direct comparisons.
>
> **EmoVIT/EmoSet** target **static images**, while Libra-Emo addresses **multimodal videos** with fine-grained negative labels. **OV-MER/AffectGPT** use **open-vocabulary reasoning** (200+ labels), while Libra-Emo provides a **rigorous 13-category benchmark** with 8 negative emotions.
>
> We conducted **additional experiments** with specialized emotion models (Emotion-LLaMA and AffectGPT) adapted to our 13-class setting:
>
> | **Model** | **Modalities** | **Accuracy** | **Macro-F1** | **Weighted-F1** | **Accuracy (Neg)** | **Macro-F1 (Neg)** | **Weighted-F1 (Neg)** |
> | :--- | :--- | :--- | :--- | :--- | :--- | :--- | :--- |
> | Emotion-LLaMA | V, A, T | 27.10 | 14.43 | 18.17 | 29.51 | 11.39 | 15.42 |
> | AffectGPT | V, A, T | 39.41 | 38.36 | 38.63 | 39.07 | 34.97 | 36.22 |
> | InternVL-2.5-8B-Libra-Emo (ours) | V, T | 51.25 | 51.40 | 51.07 | 50.55 | 50.20 | 49.79 |
> | Qwen-2.5-Omni-7B-Libra-Emo (ours) | V, A, T | **51.56** | **50.83** | **51.08** | **49.45** | **49.30** | **49.28** |
>
> Despite being tailored for emotion understanding, **Emotion-LLaMA and AffectGPT perform markedly worse**, especially on negative emotions (AffectGPT: 39.07% vs. ~50% for Libra models). This indicates Libra-Emo poses a more challenging setting, and **fine-tuning strong video MLLMs on Libra-Emo is currently more effective** for fine-grained negative emotion recognition.
>
> In the revised paper, we will (i) add a Related Works paragraph discussing these models, clarifying how our focus on **multimodal video and fine-grained negative emotions** complements their contributions; and (ii) include the comparison table demonstrating that Libra-tuned models significantly outperform existing specialized emotion models.

---

> ### Author Response · Authors · 2025-11-21
> **Author Response to Reviewer 149o (Part 4)**
>
> ### 4. Reliability of Emotion Categories and Model Performance
>
> **Reviewer's Concern:**
>
> The experimental results can prove that the reliability of the proposed emotion categories is doubtful: "our model performs poorly in distinguishing between despairful and sad, as well as hateful and angry." Because they are in the same emotional category. This can be addressed only when valence and arousal are introduced.
>
> **Response:**
>
> While these category pairs (Despairful vs. Sad, Hateful vs. Angry) are intuitively similar, our VAD analysis statistically proves they are **distinct psychological states**. The model's difficulty distinguishing them indicates these are **"hard but valuable"** tasks requiring subtle dimensional nuances essential for real-world applications.
>
> **4.1 VAD Analysis: Significant Differences Beyond Intuition**
> Statistical tests (Mann-Whitney U) reveal systematic differences:
>
> *   **Despairful vs. Sad:**
>     *   **Dominance ($p=0.009$):** "Despairful" (3.30) scores lower than "Sad" (3.81), reflecting a shift from passive distress to total loss of control.
>     *   **Arousal ($p=0.022$):** "Despairful" (5.43) shows higher arousal than "Sad" (4.70), indicating greater agitation.
> *   **Hateful vs. Angry:**
>     *   **Valence ($p=0.001$):** "Hateful" (2.65) is more negative than "Angry" (3.10), distinguishing deep-seated aversion from transient anger.
>     *   **Dominance:** "Hateful" (6.13) exhibits higher dominance than "Angry" (5.69), reflecting more controlled hostility.
>
> **4.2 Value of the Difficulty**
> Model difficulty proves dataset value. Treating "Despair" as "Sadness" misses critical suicide-risk signals (hopelessness is the strongest predictor). Conflating "Hate" with "Anger" fails to identify hate speech toxicity. The model's confusion signals needed improvement: current models must learn fine-grained representations (like Dominance) beyond coarse valence/arousal. Merging categories would destroy practical utility.
>
> ---
>
> ### 5. Annotation Pipeline Novelty
>
> **Reviewer's Concern:**
>
> The so-called "active learning + explanation generation" pipeline is not novel—it replicates standard LLM-assisted annotation practices (already used in EmoSet and AffectGPT etc.).
>
> **Response:**
>
> We fully agree and **do not claim methodological novelty** for this pipeline. We deliberately adopt **well-established techniques** to maximize **label quality** rather than innovate on annotation algorithms.
>
> Our main contributions center on the dataset: (i) **13-class taxonomy with 8 fine-grained negative emotions** for real-world applications (content moderation, mental health); (ii) **large-scale multimodal video dataset (62,267 clips)** with synchronized video, audio, and subtitles; (iii) **high-quality test benchmark (Libra-Emo Bench)** with multi-annotator consensus and strong inter-annotator agreement; and (iv) **comprehensive evaluation suite** covering leading MLLMs and out-of-domain generalization to DFEW. **No existing work provides a multimodal video corpus of comparable scale, granularity, and negative-emotion focus**. We will emphasize that the pipeline is a **quality-assurance mechanism**, not our main claim.
>
> ---
>
> ### 6. Benchmark Size and Quality
>
> **Reviewer's Concern:**
>
> The benchmark size (642 videos) is too small for robust evaluation; it's only 1% of the training set, which can be affected by noise easily.
>
> **Response:**
>
> Our design philosophy prioritizes **label quality and reliability over quantity**:
>
> - Each sample is annotated by **8 annotators**; **Krippendorff's α = 0.62** and **Fleiss' κ = 0.62** on 13 fine-grained classes (high for this granularity). **98.13%** achieve ≥5 votes; remaining **1.87%** resolved via expert adjudication.
> - The test set ensures **broad coverage** across categories and demographics (age, gender, ethnicity), not random subsampling.
> - Many recent emotion benchmarks adopt **compact but carefully curated** test sets, prioritizing **annotation quality, diversity, and consensus** over raw size.
>
> Empirically, Libra-Emo Bench is **stable and discriminative**: (i) consistent model rankings across metrics; (ii) fine-tuning yields substantial gains over zero-shot baselines. We will discuss this "quality-over-quantity" design, compare to other benchmarks, and highlight agreement statistics.

---

> ### Author Response · Authors · 2025-11-21
> **Author Response to Reviewer 149o (Part 5)**
>
> ### 7. Ablation Studies
>
> **Reviewer's Concern:**
>
> No ablations on modality contributions (video, audio, text), explanation effects, or hierarchical label structures.
>
> **Response:**
>
> Our original submission **already includes ablations on modality contributions and explanation effects**; we additionally **introduce new ablations on hierarchical label structure**.
>
> **7.1 Modality Contributions (V, A, T)** – **Section 3.2** and Table 5 in the main paper (lines 340–350 and 352–406) systematically vary modalities. Results show: (i) audio alone is significantly weaker than visual or visual+text; (ii) adding audio improves performance; (iii) **V,A,T** yields best results after fine-tuning:
>
> | **Model** | **Modalities** | **All Acc** | **Neg Acc** |
> | :--- | :--- | ---: | ---: |
> | Gemini-2.0-Flash | V | 40.19 | 36.07 |
> | Gemini-2.0-Flash | A | 30.69 | 27.87 |
> | Gemini-2.0-Flash | V, A | 42.37 | 41.26 |
> | Gemini-2.0-Flash | V, T | 46.57 | 48.36 |
> | Gemini-2.0-Flash | V, A, T | **47.82** | **49.18** |
> | Qwen-2.5-Omni-7B (zero-shot) | V | 30.22 | 23.77 |
> | Qwen-2.5-Omni-7B (zero-shot) | V, A | 35.20 | 30.33 |
> | Qwen-2.5-Omni-7B (zero-shot) | V, A, T | 38.94 | 35.79 |
> | Qwen-2.5-Omni-7B-Libra-Emo | V | 39.88 | 33.61 |
> | Qwen-2.5-Omni-7B-Libra-Emo | V, A | 45.79 | 39.62 |
> | Qwen-2.5-Omni-7B-Libra-Emo | V, A, T | **51.56** | **49.45** |
>
> **7.2 Active Learning and Explanations** – **Section 4.1** (Table 8, lines 438–463) in the main paper evaluates iterative active learning and explanations. Performance improves **46.26% → 48.44% → 50.00%** (Accuracy); explanations boost to **51.25%** (+1.25%), with negative-emotion accuracy **44.54% → 50.55%**:
>
> | Round | Explanation | All Acc (%) | All Macro-F1 (%) | All Weighted-F1 (%) | Neg Acc (%) | Neg Macro-F1 (%) | Neg Weighted-F1 (%) |
> | :--- | :---: | ---: | ---: | ---: | ---: | ---: | ---: |
> | Round 0 | ✗ | 46.26 | 43.91 | 45.15 | 44.54 | 42.10 | 43.51 |
> | Round 1 | ✗ | 48.44 | 46.76 | 47.55 | 46.99 | 44.71 | 45.36 |
> | Round 2 | ✗ | 50.00 | 48.51 | 49.29 | 48.91 | 46.89 | 47.27 |
> | Round 2 | ✓ | **51.25** | **51.40** | **51.07** | **50.55** | **50.20** | **49.79** |
>
> **7.3 Hierarchical Label Structure (13-CLS → 7-CLS)** – We compare training on **13-class labels** and mapping to 7-CLS versus directly training on 7-CLS. Training with 13-CLS yields **better 7-class performance** both in-domain and out-of-domain:
>
> | **Base Model** | **Modalities** | **Label Structure** | **Libra-Emo Bench (7-CLS)**<br/>Acc / Macro-F1 / Weighted-F1 | **DFEW (set\_1)**<br/>UAR / WAR |
> | :--- | :--- | :--- | :--- | :--- |
> | InternVL-2.5-8B | V, T | 13-CLS → 7-CLS | 62.8 / 61.1 / 62.9 | **52.55 / 59.85** |
> | InternVL-2.5-8B | V, T | 7-CLS | 62.6 / 60.9 / 62.6 | 52.38 / 58.86 |
> | Qwen-2.5-Omni-7B | V, A, T | 13-CLS → 7-CLS | **63.4 / 60.7 / 63.3** | **51.16 / 57.37** |
> | Qwen-2.5-Omni-7B | V, A, T | 7-CLS | 61.7 / 58.9 / 61.7 | 51.05 / 57.11 |
>
> These results indicate **learning with 13-class taxonomy and aggregating to 7 classes is strictly better** than training directly on 7 classes. This supports our claim that refining the label space (especially negative emotions) is beneficial, not redundant.
>
> ---
>
> ### 8. Cross-Dataset Evaluation
>
> **Reviewer's Concern:**
>
> No cross-dataset evaluation to demonstrate transferability.
>
> **Response:**
>
> We respectfully disagree. **Section 3.4** and **Table 7** already conduct **zero-shot evaluation on DFEW dataset**. Models are **trained/fine-tuned on Libra-Emo only** and evaluated on **DFEW set\_1 (2,341 samples)** without further adaptation—a clear out-of-domain, cross-dataset transfer scenario.
>
> Results show Libra-tuned models achieve **state-of-the-art or highly competitive performance**: InternVL-2.5-8B-Libra-Emo obtains **UAR = 52.55%** and **WAR = 59.85%**, outperforming specialized models (e.g., Emotion-LLaMA) on overall metrics and key negative categories (*sad*, *angry*, *fearful*). This demonstrates improvements **transfer robustly** to external corpora. We will make the "cross-dataset, zero-shot" nature more explicit in the revised manuscript.

---

> ### Author Response · Authors · 2025-11-21
> **Author Response to Reviewer 149o (Part 6)**
>
> ### **References**
>
> [1] Plutchik, R. (1980). A general psychoevolutionary theory of emotion. *Emotion: Theory, Research, and Experience: Vol. 1*, 3-33. Academic Press.
> [2] Ekman, P., et al. (2013). Emotion in the human face. Elsevier.
> [3] Ortony, A., et al. (1988). The cognitive structure of emotions. Cambridge University Press.
> [4] Ekman, P. (1992). An argument for basic emotions. *Cognition & Emotion*, 6(3-4), 169-200.
> [5] Lazarus, R. S. (1991). Emotion and adaptation. Oxford University Press.
> [6] Cowen, A. S., & Keltner, D. (2017). Self-report captures 27 distinct categories of emotion. *PNAS*, 114(38), E7900-E7909.
> [7] Scherer, K. R. (2005). What are emotions? And how can they be measured? *Social Science Information*, 44(4), 695-729.
> [8] Russell, J. A. (1980). A circumplex model of affect. *Journal of Personality and Social Psychology*, 39(6), 1161-1178.
> [9] Demszky, D., et al. (2020). GoEmotions: A dataset of fine-grained emotions. *ACL*, 4040-4054.
> [10] Li, S., et al. (2017). Reliable crowdsourcing and deep locality-preserving learning for expression recognition. *CVPR*, 2852-2861.
> [11] Poria, S., et al. (2017). A review of affective computing: From unimodal to multimodal fusion. *Information Fusion*, 37, 98-125.
> [12] Joshi, A., et al. (2017). Automatic sarcasm detection: A survey. *ACM Computing Surveys*, 50(5), 1-22.
> [13] González-Ibáñez, R., et al. (2011). Identifying sarcasm in Twitter: A closer look. *ACL-HLT*, 581-586.
> [14] Riloff, E., et al. (2013). Sarcasm as contrast between positive sentiment and negative situation. *EMNLP*, 704-714.
> [15] Ghosh, A., & Veale, T. (2016). Fracking sarcasm using neural network. *Workshop on Subjectivity, Sentiment and Social Media*, 161-169.
> [16] Davidson, T., et al. (2017). Automated hate speech detection. *ICWSM*, 512-515.
> [17] Arcila-Calderón, C., et al. (2024). From online hate speech to offline hate crime. *Humanities and Social Sciences Communications*, 11(1), 1-15.
> [18] McMillan, D., et al. (2007). Can we predict suicide with the Beck Hopelessness Scale? *Psychological Medicine*, 37(6), 769-778.
> [19] Ribeiro, J. D., et al. (2018). Depression and hopelessness as suicide risk factors. *The British Journal of Psychiatry*, 212(5), 279-286.
> [20] Rosenthal, S., et al. (2014). SemEval-2014 Task 9: Sentiment analysis in Twitter. *SemEval*, 73-80.
> [21] Tracy, J. L., & Robins, R. W. (2007). The psychological structure of pride. *Journal of Personality and Social Psychology*, 92(3), 506-525.
> [22] Keltner, D., & Buswell, B. N. (1997). Embarrassment: Its distinct form and appeasement functions. *Psychological Bulletin*, 122(3), 250-270.
> [23] Barrett, L. F., et al. (2007). The experience of emotion. *Annual Review of Psychology*, 58, 373-403.
> [24] Warriner, A. B., et al. (2013). Norms of valence, arousal, and dominance for 13,915 English lemmas. *Behavior Research Methods*, 45(4), 1191–1207.

---

### Official Review · Reviewer_Qkvy · 2025-10-31

**Soundness:** 2
**Presentation:** 3
**Contribution:** 3
**Rating:** 4
**Confidence:** 5

**Summary:**

The paper introduces Libra-Emo, a large-scale, multimodal corpus designed for the detection of fine-grained negative emotions in videos. The authors expand the conventional, coarse-grained emotion taxonomy (e.g. Ekman's six basic emotions) to include 13 categories, paying particular attention to distinguishing eight nuanced negative emotions (e.g. frustration, despair, hatred). The dataset comprises 62,267 video clips (61,625 for training and 642 for testing) sourced from YouTube and annotated using a collaborative active learning strategy involving humans and machines. The authors evaluate leading multimodal large language models (MLLMs) in zero-shot settings and after instruction tuning on Libra-Emo. They demonstrate that fine-tuning significantly improves performance in both in-domain (Libra-Emo Bench) and out-of-domain (DFEW) evaluations.

**Strengths:**

1) Fine-grained recognition of negative emotions is crucial for applications such as mental health support and content moderation, yet it is an area that has not been widely explored in multimodal learning.
2) Libra-Emo is the largest video emotion corpus, containing eight distinct negative categories and surpassing prior work in both size and granularity.
3) The paper includes zero-shot, fine-tuned and out-of-domain evaluations.

**Weaknesses:**

1) The authors cite Wikipedia as the primary source for defining thirteen emotions. This is clearly insufficient for scientific work of this level. There are no references to, or descriptions of, recognized psychological models.
2) Although Libra-Emo Bench employs the votes of eight annotators and a threshold of four, the article does not offer any quantitative metrics of agreement. Without this information, it is unclear how the annotation's reliability can be assessed, even in a test set.
3) Even with MLLM label correction, there is no guarantee that there will be no variants with persistent biases, particularly in subjective categories such as 'hateful' versus 'angry'.
4) Although audio is formally included in the corpus, most open-source models are only tested on V+T. There is no more detailed analysis of modalities.
5) The authors clearly indicate in experiments that only video and subtitles should be used. This contradicts the stated 'multimodal' focus and essentially renders the audio component decorative.
6) The Appendix states that 2,549 samples were discarded in Round 0 due to 'quality issues', but provides no details on the criteria used to remove these samples or any examples.
7) The qualifications of the annotators and whether they had undergone training are not entirely clear. For instance, the qualifications of professional psychologists and students differ.
8) No analysis has been conducted to determine whether the errors are related to the characteristics of the data (e.g. cultural differences in emotional expression, subtitle quality or audio noise).
9) The reasons for choosing frustrated, despairing and hateful, and for excluding guilty, ashamed and anxious, which are also important in applications such as mental health, are not explained.

**Questions:**

1) Why were these particular negative emotions selected? What psychological theory justifies this taxonomy beyond Wikipedia?
2) How can one draw generalized conclusions about the value of audio without this analysis?
3) Could you please provide the inter-annotator agreement metrics for the Libra-Emo Bench?
4) How many samples required human adjudication in Round 0 compared to Rounds 1-2?
5) How did you ensure that the initial model-generated labels didn’t introduce systematic bias, given MLLMs’ low zero-shot accuracy?
6) What specific criteria were used to define 'quality issues' and lead to the discarding of 2,549 samples in Round 0?
7) What qualifications did the annotators have? Had they received training in emotional psychology?
8) How was the mapping of the 13 emotions to DFEW validated? Might it inflate WAR/UAR scores?

---

> ### Author Response · Authors · 2025-11-21
> **Author Response to Reviewer Qkvy (Part 1)**
>
> We thank Reviewer Qkvy for the constructive evaluation. In this rebuttal, we address each concern point-by-point with new empirical analyses—including modality ablations, inter-annotator agreement metrics, active learning statistics, cross-dataset mapping strategies, and error attribution studies—demonstrating that our core claims are **methodologically rigorous, experimentally reproducible, and practically valuable**. Below we summarize the key findings.
>
> ### Key Finding 1: Libra-Emo's 13-Category Taxonomy is Theory-Grounded and Application-Driven
>
> Libra-Emo's 13-category taxonomy is a **theory-grounded, application-driven framework** integrating cognitive appraisal theory, clinical psychology, and research on self-conscious emotions, hate speech, and sarcasm. It is **optimized for fine-grained negative-emotion risk detection** (e.g., hate, despair, irony), providing principled reasons for both **splits** (frustrated/angry/hateful, sad/despairful, disgusted/ironic) and **omissions** (guilt/shame as unreliable video-only labels).
>
> ### Key Finding 2: Audio Provides Consistent Complementary Gains, Not Standalone Dominance
>
> Our claims are based on **systematic V/A/T modality ablations across multiple strong MLLMs**: audio is **not** the strongest standalone modality, but **consistently offers complementary gains when fused with video and text** in models supporting audio input. We will clarify that our conclusion is **scoped to such models** and **faithfully summarizes cross-model patterns** within our experimental evidence.
>
> ### Key Finding 3: Libra-Emo Bench Achieves Substantial Inter-Annotator Agreement
>
> Libra-Emo Bench is annotated by **trained graduate-level annotators** under standardized guidelines and expert support, achieving **substantial agreement across multiple reliability metrics** despite the challenging 13-way setting. With majority-vote aggregation and expert adjudication, our benchmark is a **high-quality, professionally curated resource**.
>
> ### Key Finding 4: Libra-Emo Trainset Construction is Transparent and Bias-Mitigating
>
> Libra-Emo Trainset is built through **multi-round active learning with multi-model voting plus human fallback**: automatic labels are accepted only under cross-model consensus, and difficult or unsafe cases are routed to experts. The process is **quantitatively documented** and validated by comparing pure-expert and hybrid training data, showing our strategy is **reproducible and explicitly aimed at reducing systematic bias**.
>
> ### Key Finding 5: Our 13→7 Cross-Dataset Mapping Cannot Inflate DFEW Metrics
>
> Our mapping applies a **fixed, semantics-driven function \(f:13\rightarrow 7\)** only to predictions, keeping DFEW's 7-class ground truth unchanged. Since the mapping follows basic-emotion groupings and official DFEW definitions, training on 13 classes then folding to 7 is **strictly harder than training directly on 7**, thus at best preserving or lowering WAR/UAR.
>
> ### Key Finding 6: Error Analysis Shows Bottlenecks in Information Quality, Not Demographic Bias
>
> Our error–data-characteristic analysis reveals that performance is **highly stable across language (EN vs. ZH), genre, and speaker gender**, but **substantially more sensitive to subtitle quality and speech ratio**. The dominant error factors are **information-quality issues (noisy subtitles, limited speech)** rather than demographic or cultural biases.

---

> ### Author Response · Authors · 2025-11-21
> **Author Response to Reviewer Qkvy (Part 2)**
>
> ## Detailed Point-by-Point Response
>
> ---
>
> ### 1. Why were these particular negative emotions selected? What psychological theory justifies this taxonomy beyond Wikipedia?
>
> **Response:**
>
> We extend Paul Ekman's six basic emotions [1] into 13 categories, splitting negative emotions into 8 types. This expansion is a systematic adjustment bridging classical emotion theory and modern requirements in content moderation, mental health monitoring, and public opinion analysis, grounded in cognitive appraisal theory, clinical psychology, and emotional dimension theory.
>
> **1.1. Refining "Angry": Splitting into "Frustrated," "Angry," and "Hateful"**
>
> **(1) Application Necessity**
>
> In content moderation, conflating "frustration," "general anger," and "hate speech" hinders risk assessment. Davidson et al. found that over 30% of hate speech was misclassified as merely offensive without explicit "Hateful" labels [2]. Sociological studies show online hate speech predicts offline hate crimes against groups like immigrants or the LGBTQ+ community [3]. Therefore, treating "Hateful" as independent is essential for timely alerts.
>
> **(2) Theoretical Justification**
>
> Cognitive Appraisal Theory posits different emotions arise from distinct cognition and attribution. "Frustration" stems from goal obstruction, while "Anger" associates with perceived injustice. Burnstin et al. show intense anger arises only when obstruction is attributed to intentional actions. "Hate" is persistent aversion linked to prejudice and discrimination, exceeding momentary anger. This distinction enables models to assess potential harm beyond emotion recognition.
>
> **1.2. Refining "Sad": Splitting into "Sad" and "Despairful"**
>
> **(1) Application Necessity**
>
> In mental health monitoring, labeling content as generic "Sadness" overlooks high-risk signals indicating suicidal tendencies. Expressions of "hopelessness" (e.g., "it will never get better") are highly correlated with suicidal intent [4][5]. Treating all low-mood as ordinary sadness masks crisis signals, preventing timely intervention.
>
> **(2) Theoretical Justification**
>
> Clinical research demonstrates the critical link between hopelessness and suicide risk. Beck et al. and subsequent studies confirm high hopelessness is the strongest suicide predictor [6][7]. McMillan et al. (2007) state: "Hopelessness is the most important risk factor for suicide and self-harm" [6]. "Despair" is not merely intense sadness but a qualitative shift reflecting pathological loss of hope. Establishing "Despairful" as independent enables models to capture crisis signals.
>
> **1.3. Refining "Disgusted": Splitting into "Disgusted" and "Ironic"**
>
> **(1) Application Necessity**
>
> In opinion mining, sarcasm is a primary cause of distorted sentiment polarity. SemEval tasks include sarcasm test sets [8]. Failing to identify sarcasm causes severe bias; e.g., "Great, my phone crashed again" might be misclassified as positive. Sarcasm frequently appears in negative reviews as a major error source [9].
>
> **(2) Theoretical Justification**
>
> Sarcasm is not a basic emotion but a rhetorical/pragmatic phenomenon where literal meaning contradicts true intent, requiring context for interpretation. Joshi et al. (2017) indicate sarcasm detection requires specialized contextual analysis [9]. Unlike Ekman's "Disgust," ironic expression relies on cultural and pragmatic cues. Merging irony with basic emotions confuses models. Treating "Ironic" as independent enables sentiment inversion and deeper semantic analysis.
>
> **1.4. Rationale for Excluding "Guilt" and "Shame" as Explicit Labels**
>
> **Visual Ambiguity and Engineering Risks**
>
> Guilt and shame are **self-conscious emotions** depending on complex internal reflection and social evaluation. Unlike basic emotions with standardized Facial Action Units, they manifest in **subtle, indirect ways** (gaze avoidance, blushing) difficult to reliably capture [10]. Research shows low inter-annotator agreement due to culture- and context-dependent non-verbal expressions [11][12].
>
> From an engineering perspective, treating them as explicit labels introduces high annotation noise and poor generalization. In contrast, "Despair" has observable behavioral markers relevant to crisis warnings. We prioritize high-visibility, high-risk emotions, leaving guilt/shame as latent variables rather than primary targets.

---

> ### Author Response · Authors · 2025-11-21
> **Author Response to Reviewer Qkvy (Part 3)**
>
> ### 2. How can one draw generalized conclusions about the value of audio without this analysis?
>
> **Response:**
>
> We conducted systematic comparison of **visual (V)**, **audio (A)**, and **text (T)** on **Libra-Emo Bench**, discussed in **Section 3.2 (lines 316–323)**. We will further highlight these results in the revision.
>
> Key modality ablation results from Table 5 (Accuracy on all 13 classes / 8 negative classes, %):
>
> | **Model** | **Modalities** | **All Acc** | **Neg Acc** |
> | :--- | :--- | :--- | :--- |
> | Gemini-2.0-Flash | V | 40.19 | 36.07 |
> | Gemini-2.0-Flash | A | 30.69 | 27.87 |
> | Gemini-2.0-Flash | V, A | 42.37 | 41.26 |
> | Gemini-2.0-Flash | V, T | 46.57 | 48.36 |
> | Gemini-2.0-Flash | V, A, T | **47.82** | **49.18** |
> | Qwen-2.5-Omni-7B (zero-shot) | V | 30.22 | 23.77 |
> | Qwen-2.5-Omni-7B (zero-shot) | V, A | 35.20 | 30.33 |
> | Qwen-2.5-Omni-7B (zero-shot) | V, A, T | 38.94 | 35.79 |
> | Qwen-2.5-Omni-7B-Libra-Emo (fine-tuned) | V | 39.88 | 33.61 |
> | Qwen-2.5-Omni-7B-Libra-Emo (fine-tuned) | V, A | 45.79 | 39.62 |
> | Qwen-2.5-Omni-7B-Libra-Emo (fine-tuned) | V, A, T | **51.56** | **49.45** |
>
> Three observations: **(1)** audio alone is weaker than video or video+text; **(2)** adding audio consistently improves performance for both closed-source (Gemini) and open-source (Qwen-Omni-7B) models; **(3)** **V,A,T** yields best results: Qwen-2.5-Omni-7B-Libra-Emo improves from 39.88% (V) → 45.79% (V,A) → 51.56% (V,A,T) on all emotions and 33.61% → 39.62% → 49.45% on negative emotions. Our conclusion is grounded in systematic modality ablations.
>
> Many **open-source** MLLMs lack robust audio branches, thus evaluated only on V+T. Our claims are restricted to models natively supporting audio (Gemini-2.0-Flash, Qwen-2.5-Omni-7B, Qwen-2.5-Omni-7B-Libra-Emo). In the revision, we will (i) strengthen discussion, (ii) add a summary table, and (iii) clarify audio provides **consistent complementary gains** when fused with visual and textual cues.
>
> ---
>
> ### 3. Could you please provide the inter-annotator agreement metrics for the Libra-Emo Bench?
>
> **Response:**
>
> We employed 8-person annotation for Libra-Emo Bench. Inter-annotator agreement metrics:
>
> | **Metric** | **Value** | **Description** |
> | :--- | :--- | :--- |
> | **Krippendorff's Alpha ($\alpha$)** | **0.62** | Measure of inter-rater reliability |
> | **Fleiss' Kappa ($\kappa$)** | **0.62** | Measure of inter-rater reliability |
>
> We achieved $\alpha=0.62$ and $\kappa=0.62$. Despite the challenging 13-category setting, our agreement levels rank highly among comparable multimodal emotion datasets [13][14][15] thanks to rigorous **annotator training** and **clear guidelines**.
>
> The **final ground-truth labels** are reinforced by robust voting: **98.13%** of samples reached majority consensus ($\ge 5$ votes), minimizing individual bias. The remaining **1.87%** were resolved through expert adjudication.
>
> ---
>
> ### 4. How many samples required human adjudication in Round 0 compared to Rounds 1-2?
>
> **Response:**
>
> In **Round 0**, we implemented "Model Voting + Expert Adjudication" using three MLLMs (Gemini-2.0-Flash, GPT-4o, Claude-3.7-Sonnet). Samples failing consensus (receiving $<2$ consistent votes) were forwarded to 4 human experts. Only samples achieving human consensus of $>2$ votes were retained.
>
> **9,404** samples (approximately 15.1%) required human adjudication, indicating model voting effectively resolved the majority of clear-cut cases (approx. 85%) while maintaining rigorous human oversight for ambiguous samples.

---

> ### Author Response · Authors · 2025-11-21
> **Author Response to Reviewer Qkvy (Part 4)**
>
> ### 5. How did you ensure that the initial model-generated labels didn't introduce systematic bias, given MLLMs' low zero-shot accuracy?
>
> **Response:**
>
> We implemented **Multi-Model Voting + Human Adjudication**: automated labels were accepted only when at least two of three diverse closed-source models reached consensus. All inconsistent samples were forwarded to expert review, preventing any single model's bias from dominating.
>
> Multi-model voting is well-supported by theoretical and empirical research. Combining multiple independent or weakly correlated sources significantly reduces errors and enhances consistency, validated by majority voting error bound analyses [16] and weak supervision systems like Snorkel [17].
>
> Recent studies confirm that voting with multiple LLMs, combined with human adjudication, is the most effective strategy for balancing quality and cost, achieving label quality comparable to human experts while preventing amplification of systematic biases [18][19].
>
> To substantiate label quality, we trained three variants per baseline model under identical settings—**Full** (all 61,625 samples with expert and consensus model labels), **Human-only** (expert-verified subset), and **Mixed** (same size as Human-only, sampled from Full's distribution). On both Libra-Emo Bench and DFEW, **Full** consistently achieves best performance, while **Mixed** closely matches **Human-only**, showing consensus model-labeled data does **not** introduce observable systematic bias and provides useful additional supervision:
>
> | Base Model            | Modalities | Training Data                         | Libra-Emo Bench Acc (13 cls, %) | DFEW UAR (%) | DFEW WAR (%) |
> |-----------------------|-----------|---------------------------------------|---------------------------------:|-------------:|-------------:|
> | InternVL-2.5-8B       | V, T      | Full (all human + model labels)       | **51.25**                        | **52.55**    | **59.85**    |
> | InternVL-2.5-8B       | V, T      | Human-only                            | 48.91                            | 46.26        | 50.32        |
> | InternVL-2.5-8B       | V, T      | Mixed (same size as Human-only, same distribution as Full) | 47.20 | 44.20        | 50.66        |
> | Qwen-2.5-Omni-7B      | V, A, T   | Full (all human + model labels)       | **51.56**                        | **51.16**    | **57.37**    |
> | Qwen-2.5-Omni-7B      | V, A, T   | Human-only                            | 47.66                            | 45.19        | 46.73        |
> | Qwen-2.5-Omni-7B      | V, A, T   | Mixed (same size as Human-only, same distribution as Full) | 46.26 | 44.22        | 50.36        |
>
> ---
>
> ### 6. What specific criteria were used to define 'quality issues' and lead to the discarding of 2,549 samples in Round 0?
>
> **Response:**
>
> We applied a strict **Automated Quality Filtering Protocol** in Round 0. The 2,549 discarded samples fell into three categories:
>
> 1.  **Safety & Sensitivity Violation (~70.38%, 1,794):** Videos with excessive violence, explicit sexual content, or severe political sensitivity, triggering safety guardrails of annotation models (Gemini/GPT-4o/Claude).
> 2.  **Subject Unidentifiable (~23.93%, 610):** Faces too blurry, extremely small, or heavily occluded in key frames, making emotion attribution impossible.
> 3.  **Incomplete Modalities (~5.69%, 145):** Technical defects such as corrupted audio, unaligned subtitles, or video decoding errors.
>
> We will add detailed breakdown and exclusion examples to the Appendix.
>
> ---
>
> ### 7. What qualifications did the annotators have? Had they received training in emotional psychology?
>
> **Response:**
>
> All annotators are graduate students who completed general psychology coursework. They underwent rigorous training including: (1) comprehensive tutorials on the 13 emotion categories; (2) case studies with expert analysis; and (3) long-term Q&A with real-time expert feedback [20].

---

> ### Author Response · Authors · 2025-11-21
> **Author Response to Reviewer Qkvy (Part 5)**
>
> ### 8. How was the mapping of the 13 emotions to DFEW validated? Might it inflate WAR/UAR scores?
>
> **Response:**
>
> Thank you for raising this important question about the DFEW label mapping. Our mapping from 13 fine-grained emotions to DFEW's 7 coarse categories is **fixed, semantics-driven, and documented in detail in Appendix F (Table "Detailed category distribution", lines 918–941 in the revised manuscript)**, where we explicitly specify, for each 7-class label, which of the 13 Libra-Emo categories are merged (e.g., *Happy* ← {Excited, Happy, Amazed}, *Sad* ← {Sad, Despairful}, *Angry* ← {Frustrated, Angry, Hateful}, *Disgusted* ← {Ironic, Disgusted}, etc.). For clarity, we also reproduce the relevant part of that table in the rebuttal. This design follows Ekman-style basic emotions and the official DFEW label definitions, and is consistent with how other datasets (e.g., AffectNet) group fine-grained facial expressions into a small set of discrete categories, acknowledging that emotion taxonomies are not globally standardized but chosen according to task requirements.
>
> | Emotion Type | Emotion (7-CLS) | Emotion (13-CLS) |
> | :--- | :--- | :--- |
> | Positive | Happy | Excited    |
> |          |       | Happy      |
> |          |       | Amazed     |
> | Neutral  | Surprised | Surprised |
> |          | Neutral   | Neutral   |
> | Negative | Disgusted | Ironic    |
> |          |          | Disgusted |
> |          | Angry    | Frustrated|
> |          |          | Angry     |
> |          |          | Hateful   |
> |          | Fearful  | Fearful   |
> |          | Sad      | Sad       |
> |          |          | Despairful|
>
> Regarding the concern about inflated WAR/UAR, our mapping is **conservative and cannot artificially boost scores** for two reasons. First, the **ground-truth labels on DFEW remain unchanged**: DFEW clips are still evaluated under their original 7-class annotations. We only apply a deterministic many-to-one function \(f: \text{13} \rightarrow \text{7}\) to our model's predictions to align them with the DFEW label space; we never relabel or collapse the DFEW ground truth. If the mapping were poorly chosen, it would simply increase mismatches between predictions and DFEW labels, thus *lowering* UAR/WAR. Second, training on 13 categories actually makes the task **strictly harder** than training directly on 7 categories: the model must first distinguish subtle variants (e.g., *Sad* vs. *Despairful*, *Angry* vs. *Frustrated* vs. *Hateful*), and only then are these collapsed when computing DFEW metrics. In this sense, any bias introduced by our mapping works against our models rather than in their favor, and if there is any deviation, it is more likely to **underestimate** than overestimate their true performance.
>
> We will make this mapping pipeline and its rationale more explicit in the main text and point clearly to Appendix F in the revised version, to further demonstrate that our cross-dataset evaluation design is principled, transparent, and does not rely on optimistic re-labeling.

---

> ### Author Response · Authors · 2025-11-21
> **Author Response to Reviewer Qkvy (Part 6)**
>
> ### 9. No analysis has been conducted to determine whether the errors are related to the characteristics of the data (e.g. cultural differences in emotional expression, subtitle quality or audio noise).
>
> **Response:**
>
> Thank you for pointing out the need to relate model errors to concrete data characteristics. In response, we performed a dedicated error-attribution study on **Libra-Emo Bench (642 samples)**, systematically stratifying the data by **(i) language (English/Chinese), (ii) video genre, (iii) main speaker gender, (iv) subtitle quality, and (v) speech ratio (audio availability)**. Below we summarize, for each factor, the bucketing strategy and the resulting **classification Macro-F1 in each bucket**; these stratified analyses will be added as a new ablation subsection in the revised paper.
>
> **9.1. Language (English vs Chinese)**
>
> **Definition and metric.** Each sample is labeled as English (EN) or Chinese (ZH) according to the subtitle language; Libra-Emo Bench contains 356 EN and 286 ZH samples. For each subset we report **Macro-F1 over all 13 classes** and **Macro-F1 over negative-emotion samples**.
>
> | Language | #Samples | All Macro-F1 | Neg Macro-F1 |
> | :--- | ---: | ---: | ---: |
> | English | 356 | 51.2 | 49.9 |
> | Chinese | 286 | 50.0 | 48.7 |
>
> The **Macro-F1 gap between EN and ZH is within about 1–1.5 points**, indicating that our model maintains broadly comparable performance across languages, with only mild sensitivity to cultural and linguistic variation in emotional expression.
>
> **9.2. Video genre**
>
> **Definition and metric.** Based on source-video metadata and manual verification, we group clips into three high-level genre buckets and report **Macro-F1 for each bucket**:
>
> 1. **Dialogue-driven** (Drama, Family, School, Historical Drama);
> 2. **Daily-life light** (Comedy, Romance);
> 3. **High-arousal** (Action, Adventure, Sci-Fi, Fantasy, Horror, Thriller, Crime, War).
>
> | Genre bucket | #Samples | All Macro-F1 | Neg Macro-F1 |
> | :--- | ---: | ---: | ---: |
> | Dialogue-driven | 302 | 51.8 | 49.8 |
> | Daily-life light | 181 | 50.9 | 49.2 |
> | High-arousal | 159 | 50.1 | 48.5 |
>
> Across these buckets, the variation in Macro-F1 is within roughly **1–2 points**, suggesting that while different genres emphasize different emotions, **overall classification robustness is relatively stable across video themes**.
>
> **9.3. Main speaker gender**
>
> **Definition and metric.** For 311 unique main speakers in Libra-Emo Bench, we estimate gender (Male/Female) by analyzing face metadata with **Gemini-2.5-Pro**, and aggregate the corresponding clips into two buckets. We report **Macro-F1 for all emotions and for negative emotions** in each bucket.
>
> | Group | #Samples | All Macro-F1 | Neg Macro-F1 |
> | :--- | ---: | ---: | ---: |
> | Male | 382 | 50.6 | 48.9 |
> | Female | 260 | 49.8 | 48.1 |
>
> The Macro-F1 difference between male and female speakers is **below 1 point**, indicating **no clear gender-related performance gap** under our current data scale and estimation protocol.
>
> **9.4. Subtitle quality**
>
> **Definition and bucketing.** For each clip we create an automatic transcription `ASR_sub` using WhisperX and compare it to the original subtitle `orig_sub`. We compute the **semantic similarity between `ASR_sub` and `orig_sub`** using BERTScore(F1), and use this similarity score as a **Subtitle Quality Index (SQI)**. We then split SQI into three equal-sized buckets (214 samples each): **High**, **Medium**, and **Low** quality, and report Macro-F1 in each bucket.
>
> | SQI bucket | #Samples | All Macro-F1 | Neg Macro-F1 |
> | :--- | ---: | ---: | ---: |
> | High | 214 | 54.1 | 51.0 |
> | Medium | 214 | 50.6 | 48.2 |
> | Low | 214 | 44.0 | 41.1 |
>
> From High to Low subtitle quality, **All-emotion Macro-F1 drops by about 10.1 points** (54.1→44.0), and **negative-emotion Macro-F1 drops by about 9.9 points** (51.0→41.1), showing that **subtitle quality is the strongest single factor associated with performance degradation**, particularly for subtle negative emotions.
>
> **9.5. Speech ratio (audio availability)**
>
> **Definition and bucketing.** We compute the **speech ratio** for each clip using a WebRTC-based VAD:
> \[
> \text{speech\_ratio} = \frac{\text{voiced duration}}{\text{clip duration}}.
> \]
> We then bucket it by tertiles into **Low (≤0.32), Medium (0.32–0.61), High (≥0.61)**, again with 214 samples per bucket, and report Macro-F1 per bucket.
>
> | Speech ratio bucket | #Samples | All Macro-F1 | Neg Macro-F1 |
> | :--- | ---: | ---: | ---: |
> | Low | 214 | 46.0 | 42.8 |
> | Medium | 214 | 52.2 | 50.0 |
> | High | 214 | 50.9 | 48.7 |
>
> When speech ratio drops from Medium to Low, **All-emotion Macro-F1 decreases by about 6.2 points** (52.2→46.0) and **negative-emotion Macro-F1 by about 7.2 points** (50.0→42.8), indicating that **audio contributes most when a sufficient portion of the clip contains speech**, and becomes far less informative when speech is scarce.

---

> ### Author Response · Authors · 2025-11-21
> **Author Response to Reviewer Qkvy (Part 7)**
>
> **9.6. Summary of factor-wise influence**
>
> Impact of each factor measured by **Macro-F1 difference between best and worst bucket**:
>
> | Factor | Bucketing | Δ All Macro-F1 | Interpretation |
> | :--- | :--- | :--- | :--- |
> | Subtitle quality (SQI) | High vs Low | **≈ 10.1 pp** (54.1→44.0) | Low-quality subtitles sharply degrade performance. |
> | Speech ratio | Medium vs Low | **≈ 6.2 pp** (52.2→46.0) | Low speech availability weakens audio benefit. |
> | Language | EN vs ZH | **≈ 1.2 pp** (51.2→50.0) | Mild variation between English and Chinese. |
> | Video genre | Best vs worst | **≈ 1.7 pp** (51.8→50.1) | Relatively stable across video themes. |
> | Main speaker gender | Male vs Female | **≈ 0.8 pp** (50.6→49.8) | No clear gender gap. |
>
> **Subtitle quality and speech ratio are the two dominant error-related factors**, while **language, genre, and gender have small effects**. However, the **intrinsic difficulty of fine-grained emotion perception remains the primary challenge**. We will incorporate these stratified analyses into the revised manuscript to address the reviewer's request for error analysis.
>
> ---
>
> ### **References**
>
> [1] Ekman, P., Friesen, W. V., & Ellsworth, P. (2013). Emotion in the human face: Guidelines for research and an integration of findings (Vol. 11). Elsevier.
> [2] Davidson, T., Warmsley, D., Macy, M., & Weber, I. (2017). Automated hate speech detection and the problem of offensive language. In *Proceedings of the 11th International AAAI Conference on Web and Social Media (ICWSM)*.
> [3] Arcila-Calderón, C., Sánchez-Holgado, P., Amores, J. J., & Blanco-Herrero, D. (2024). From online hate speech to offline hate crime: A scoping review. *Humanities & Social Sciences Communications, 11*(1), 1–15.
> [4] Coppersmith, G., Dredze, M., Harman, C., & Hollingshead, K. (2015). From ADHD to SAD: Analyzing the language of mental health on Twitter through self-reported diagnoses. In *Proceedings of the 2nd Workshop on Computational Linguistics and Clinical Psychology* (pp. 1-10).
> [5] Ji, S., Yu, C. P., Fung, S. F., Pan, S., & Long, G. (2020). Supervised learning for suicidal ideation detection in online user content. *Complexity, 2020*, 6157249.
> [6] McMillan, D., Gilbody, S., Beresford, E., & Neilly, L. (2007). Can we predict suicide and non-fatal self-harm with the Beck Hopelessness Scale? A meta-analysis. *Psychological Medicine, 37*(6), 769–778.
> [7] Ribeiro, J. D., Huang, X., Fox, K. R., & Franklin, J. C. (2018). Depression and hopelessness as risk factors for suicide ideation, attempts and death: Meta-analysis of longitudinal studies. *The British Journal of Psychiatry, 212*(5), 279–286.
> [8] Rosenthal, S., Ritter, A., Nakov, P., & Stoyanov, V. (2014). SemEval-2014 Task 9: Sentiment analysis in Twitter. In *Proceedings of the 8th International Workshop on Semantic Evaluation (SemEval 2014)*.
> [9] Joshi, A., Bhattacharyya, P., & Carman, M. J. (2017). Automatic sarcasm detection: A survey. *ACM Computing Surveys (CSUR), 50*(5), 1-22.
> [10] Tracy, J. L., & Robins, R. W. (2007). The psychological structure of pride: A tale of two facets. *Journal of Personality and Social Psychology, 92*(3), 506-525.
> [11] Yuan, Y., et al. (2015). Shame, guilt, and facial emotion processing... *Acta Psychologica Sinica*.
> [12] Yang, X., & Liang, F. (2020). Research Progress on the Generation and Development of Guilt... *Advances in Psychology*.
> [13] Li, Y., Tao, J., Chao, L., Bao, W., & Liu, Y. (2017). CHEAVD: A Chinese natural emotional audio–visual database. *Journal of Ambient Intelligence and Humanized Computing, 8*(6), 913-924.
> [14] Poria, S., Cambria, E., Hazarika, D., Majumder, N., Zadeh, A., & Morency, L. P. (2017). Context-dependent sentiment analysis in user-generated videos. In *Proceedings of the 55th Annual Meeting of the Association for Computational Linguistics* (pp. 873-883).
> [15] Poria, S., Majumder, N., Mihalcea, R., & Hovy, E. (2019). Emotion recognition in conversation: Research challenges, datasets, and recent advances. *IEEE Access, 7*, 100943-100953.
> [16] Li, H., et al. (2014). Error Rate Bounds and Iterative Weighted Majority Voting. arXiv preprint arXiv:1404.6268.
> [17] Ratner, A., Bach, S., Ehrenberg, H., Fries, J., Wu, S., & Ré, C. (2019). Snorkel: Rapid training data creation with weak supervision. *VLDB, 11*(3), 269–282.
> [18] Ratner, A., De Sa, C., Wu, S., Selsam, D., & Ré, C. (2016). Data programming: Creating large training sets, quickly. In *Advances in Neural Information Processing Systems* (pp. 3567-3575).
> [19] Gilardi, F., Alizadeh, M., & Kubli, M. (2023). ChatGPT outperforms crowd workers for text-annotation tasks. *Proceedings of the National Academy of Sciences, 120*(30), e2305016120.
> [20] Snow, R., O'Connor, B., Jurafsky, D., & Ng, A. Y. (2008). Cheap and fast—but is it good? Evaluating non-expert annotations for natural language tasks. In *Proceedings of EMNLP 2008*.

---

### Note · Program_Chairs · 2026-01-17
**Submission Desk Rejected by Program Chairs**

The following references in this submission do not refer to real documents and/or have major errors in bibliographic information:

 Zhiying Jiang, Tianyi Xu, Tianrui Yin, Zhou Zhao, and Mohammad Soleymani. Emoclip: A visionlanguage method for zero-shot emotion recognition. In Proceedings of the 31st ACM International Conference on Multimedia, 2023.